# Stochastic coalescence in Lagrangian cloud microphysics

Piotr Dziekan and Hanna Pawlowska

Institute of Geophysics, Faculty of Physics, University of Warsaw, Poland

*Correspondence to:* P. Dziekan (pdziekan@igf.fuw.edu.pl)

**Abstract.** Stochasticity of the collisional growth of cloud droplets is studied using the super-droplet method (SDM) of Shima et al. (2009). Statistics are calculated from ensembles of simulations of collision-coalescence in a single well-mixed cell. The SDM is compared with direct numerical simulations and the master equation. It is argued that SDM simulations in which one computational droplet represents one real droplet are at the same level of precision as the master equation. Such sim-
ulations are used to study fluctuations in the autoconversion time, the sol-gel transition and the growth rate of lucky droplets, which is compared with a theoretical prediction. Size of the coalescence cell is found to strongly affect system behavior. In small cells, correlations in droplet sizes and droplet depletion slow down rain formation. In large cells, collisions between rain drops are more frequent and this also can slow down rain formation. The increase in the rate of collision between rain drops may be an artefact caused by assuming a too large well-mixed volume. The highest ratio of rain water to cloud water is found
in cells of intermediate sizes. Next, we use these precise simulations to determine validity of more approximate methods: the Smoluchowski equation and the SDM with mulitplicities greater than 1. In the latter, we determine how many computational droplets are necessary to correctly model the expected number and the standard deviation of autoconversion time. Maximal size of a volume that is turbulently well-mixed with respect to coalescence is estimated at $V_{\mathrm{mix}} = 1.5 \cdot 10^{-2}$ cm$^3$. The Smoluchowski equation is not valid in such small volumes. It is argued that larger volumes can be considered approximately well-mixed, but such approximation needs to be supported by a comparison with fine-grid simulations that resolve droplet motion.

## 1 Introduction

Coalescence of hydrometeors is commonly modeled using the Smoluchowski equation (Smoluchowski, 1916), often also called the stochastic coalescence equation. It is a mean-field equation that can be derived from the more fundamental stochastic description by neglecting correlations between numbers of droplets of different sizes (Gillespie, 1972; Bayewitz et al., 1974). These correlations are especially important in small volumes and neglecting them can lead to unphysical behavior. For example, when a single drop contains majority of water in a coalescence cell (gelation), the Smoluchowski equation does not conserve mass for some coalescence kernels (Leyvraz, 2003).

Another limitation of the Smoluchowski equation is that it describes evolution only of the expected number of droplets of given size. It does not contain information about fluctuations around this number, which are suspected to be crucial for precipitation onset (Telford, 1955; Scott, 1967; Marcus, 1968). Rate of collisions between droplets depends on their sizes. Small droplets rarely collide with each other, because they are repelled by disturbance flow induced by their settling. Once a

droplet reaches a threshold size, it becomes more efficient at collecting smaller droplets. The mean time for a droplet to reach the threshold size is long, but some lucky droplets could reach it much sooner through a series of unlikely collisions. Then they grow quickly, resulting in a sooner onset of precipitation. This effect cannot be described using the Smoluchowski equation.

Moreover, although the Smoluchowski equation can be written for the discrete number of droplets of given size, it is more often used for droplet concentrations. This adds an additional assumption that the coalescence volume is large, somewhat in agreement with neglectng fluctuations in number of collisions and correlations in droplet numbers (Gillespie, 1972).

A number of methods alternative to the Smoluchowski equation exist. They are capable of addressing stochastic coalescence, but have some shortcomings that make their use in large-scale cloud simulations impossible. The most accurate one is the direct numerical simulation (DNS). In DNS, trajectories of droplets are simulated explicitly and collisions occur when they come in contact. The downside of DNS is that it is computationally extremely demanding. Running large ensemble of simulations from which statistics could be obtained would take prohibitively long time. An alternative approach is to use a master equation (Gillespie, 1972). It describes temporal evolution of probability of observing a given number of particles of a given size. Collisions are allowed between all particles in some coalescence volume and are assumed to be Markovian, i.e. they only depend on the instantaneous state of the system and not on its history. This can only be justified if the volume is well-mixed, i.e. if droplets are randomly redistributed within the volume after each collision. It is worth to note that DNS does not require such assumptions, so it reproduces correlations between positions and sizes of droplets. The master equation was analytically solved only for monodisperse initial conditions with simple coalescence kernels (Bayewitz et al., 1974; Tanaka and Nakazawa, 1993). A more general form of the Bayewitz equation is given in Wang et al. (2006), but cannot be solved for any realistic coalescence kernel. Solving the master equation numerically is extremely difficult due to a huge phase space to be considered. Recently, Alfonso (2015) developed a method to solve the master equation numerically, but was only able to apply the method to a system of up to 40 droplets (Alfonso and Raga, 2017). Alternatively, the stochastic simulation algorithm (SSA) (Gillespie, 1975; Seesselberg et al., 1996) can be used to model single trajectory obeying the master equation, but obtaining large enough statistics would require very long computations.

Several Lagrangian methods have been developed to model cloud microphysics (Andrejczuk et al., 2008; Shima et al., 2009; Sölch and Kärcher, 2010; Riechelmann et al., 2012). Their common point is that they explicitly model microphysical processes on a small population of computational particles, each representing a large number of real particles. We will refer to these computational particles as super-droplets (SDs). The words "droplets" and "drops" are reserved for real hydrometeors. A thorough comparison of coalescence algorithms from Lagrangian methods was done by Unterstrasser et al. (2017). It lead to the conclusion that the method of Shima et al. (2009) "yields the best results and is the only algorithm that can cope with all tested kernels". It was also found to be optimal in DNS tests (Li et al., 2017). In the light of these results, we choose to use the coalescence algorithm of Shima et al. (2009) in this work. Throughout the paper, by the name "super-droplet method" we refer to this specific algorithm and any conclusions regarding SDM are valid only for the Shima SDM. The Shima et al. (2009) algorithm is not based on the Smoluchowski equation, but, similarly to the master equation, on the assumption that the volume is well-mixed. The algorithm introduces some simplifications that may increase the scale of fluctuations in the number of collisions, as described in Sec. 2. These simplifications are not necessary in the limiting case of a single computational particle

representing a single real particle, what we call "one-to-one" simulations. Then, the Shima et al. (2009) algorithm should be equivalent to the SSA, i.e. it should produce a single realization in agreement with the master equation. To show that this is true, we compare the Shima et al. (2009) algorithm with the master equation and the SSA in Sec. 3. We also compare it with the more fundamental DNS approach in Sec. 4. Once the "one-to-one" approach is shown to be at the same level of precision

as the master equation, we use it to study some physical processes that are related to the stochastic nature of coalescence. The way the sol-gel transition time changes with cell size is studied in Sec. 3 and in Sec. 6, we quantify how quickly the luckiest cloud droplets become rain drops. In addition, we use the "one-to-one" approach to validate more approximate methods. The Shima et al. (2009) algorithm with multiplicities greater than 1 is studied in Sec. 4. We determine how many computational particles are required to obtain the correct mean autoconversion time and correct fluctuations in the autoconversion time.

Next, in Sec. 5, we determine how large a cell has to be for the Smoluchowski equation to correctly represent the rate of rain formation. Throughout the paper we observe that evolution of the droplet size spectrum strongly depends on size of the coalescence cell. The size of a well-mixed air volume is estimated in Sec. 7 and some implications for cloud simulations are discussed in Sec. 8.

## 2   The super-droplet method

In this Section we present how collision-coalescence is handled in the super-droplet method. Further information about SDM can be found in Shima et al. (2009). Consider coalescence of water droplets in a well-mixed volume $V$. Other processes, like water condensation and evaporation, are not included. Thanks to the assumption that the volume is well-mixed, all droplets within the volume can collide with each other, independently of their positions (Gillespie, 1972). Therefore droplet motion does not have to be explicitly modeled and droplet coalescence can be calculated in a stochastic manner, as it is done in the

master equation. Consider two randomly selected droplets $i$ and $j$. Probability that they collide during the timestep $\Delta t$ is $P(r_i, r_j) = K(r_i, r_j)\Delta t/V$, where $r_i$ and $r_j$ are their radii and $K$ is the coalescence kernel. We use gravitational coalescence kernels, so the effect of turbulence on coalescence is not studied.

At the heart of the super-droplet method is the idea that many droplets with same properties within a well-mixed volume can be represented by a single computational entity, called the super-droplet (SD). As we are interested only in droplet coalescence

within a single cell, it is sufficient if SDs are characterized by two parameters: radius $r$ and multiplicity $\xi$, which is the number of real droplets that a SD represents. Only integer multiplicities are allowed. In the algorithm of Shima et al. (2009), two simplifications are made that may affect the amplitude of fluctuations in the number of collisions. The first simplification is that SDs collide in an "all-or-nothing" manner. If a collision happens, each real droplet represented by the SD with lower multiplicity collides with a single droplet represented by the SD with higher multiplicity. If the $i$-th and $j$-th SDs collide, their

parameters are updated to:

$$\xi'_j = \xi_j, \quad \xi'_i = \xi_i - \xi_j$$
$$r'_j = (r_i^3 + r_j^3)^{1/3}, \quad r'_i = r_i, \tag{1}$$

where primes denote post-collisional values and we assume $\xi_j \leq \xi_i$. Intuitively, one would expect the "all-or-nothing" procedure to lead to larger fluctuations than in a real system, because the number of collision trials is artificially reduced. The second simplification, that we will refer to as "linear sampling", is that instead of considering all $N(N-1)/2$ collision pairs, only $\lfloor N/2 \rfloor$ non-overlapping pairs are randomly selected. $N$ is the number of SDs in the coalescence cell and $\lfloor x \rfloor$ stands for the largest integer equal to, or smaller than $x$. To keep the expected number of collisions equal to the real one, coalescence probabilities are scaled up. Probability of coalescence of two SDs $i$ and $j$ that belong to the same collision pair is $P_{SD}(r_i, r_j, \xi_i, \xi_j) = \max(\xi_i, \xi_j) P(r_i, r_j)(N(N-1)/2)/\lfloor N/2 \rfloor$ (Shima et al., 2009). Real droplets represented by the same SD cannot collide with each other, because they have the same sedimentation velocities.

We will perform two types of simulations. In the "one-to-one" simulations, all SDs have multiplicity $\xi = 1$. That way the "all-or-nothing" simplification is removed. $N_0$ super-droplets are initialized by randomly drawing radii from the assumed initial droplet size distribution, where $N_0$ is the initial number of real droplets in a cell. Coalescence causes one of the SDs to be discarded. Unlike in the original method of Shima et al. (2009), timestep length is adapted at each timestep to ensure that none of the collision pairs has coalescence probability greater than 1. This approach is similar to the Direct Simulation Monte Carlo method used in diluted gas dynamics (Bird, 1994). In Sec. 3 we show that the "one-to-one" method is in agreement with the master equation.

The second type of simulations, in which number of SDs remains constant (with rare exceptions), is closer to the ideas of Shima et al. (2009). We will refer to it as the "constant SD" simulations. In this type of simulations, the number of SDs is prescribed as $N_{SD}$ and SDs have different multiplicities. Typically, $N_{SD}$ is much smaller than $N_0$. We use a novel algorithm for initialising the radii and multiplicities of SDs. The aim of this algorithm is to avoid large differences in the initial droplet size distributions between realizations. Super-droplet radii are not completely randomly drawn from the assumed distribution as in the "one-to-one" simulations. Instead, the assumed distribution is divided into $N_{SD}$ bins and the radius of a single SD is randomly selected within each bin. The bins have equal sizes on a logarithmic scale. Consider an initial droplet size distribution $n(\ln(r))$. We employ a notation in which we omit the division of radius by unit of length whenever the logarithm of radius is taken, i.e. $\ln(r)$ stands for $\ln(r/\mu m)$. Concentration of droplets with radii in the range from $r$ to $r+dr$ is $n|_{r,r+dr} = n(\ln(r)) \mathrm{d}\ln(r)$. The first step of the initialization is to find the largest and the smallest initial super-droplet radius, $r_{\max}$ and $r_{\min}$. They are found iteratively, starting with $r_{\min} = 10^{-9}$ m and $r_{\max} = 10^{-3}$ m. We require that they satisfy the condition

$$n(\ln(r_e))\Delta l_r V \geq 1, \tag{2}$$

where $r_e$ is either $r_{\max}$ or $r_{\min}$ and $\Delta l_r = (\ln(r_{\max}) - \ln(r_{\min}))/N_{SD}$. In each iteration, if $r_{\min}$ ($r_{\max}$) does not satisfy (2), it is increased (decreased) by 1%. Once $r_{\min}$ and $r_{\max}$ are found, $N_{SD}$ super-droplets are created. Radius of the $i$-th SD is initialised by randomly selecting $\ln(r_i)$ in the range $(\ln(r_{\min}) + (i-1)\Delta l_r, \ln(r_{\min}) + i\Delta l_r]$. Initial multiplicity of the $i$-th SD is $\xi_i = \lfloor n(\ln(r_i))\Delta l_r V + 0.5 \rfloor$. Please note that increasing $N_{SD}$ causes $\Delta l_r$ to decrease and this in turn gives relatively large values of $r_{\min}$ and relatively small values of $r_{\max}$. It means that this initialisation procedure does not represent well tails of the distribution, especially for large $N_{SD}$. It also means that the "constant SD" initialisation with $N_{SD} = N_0$ is not equivalent to the "one-to-one" initialisation. Since the large tail is important for coalescence, we add

$\lfloor \int_{\ln(r_{\max})}^{\infty} n(\ln(r)) \, \mathrm{d} \ln(r) V + 0.5 \rfloor$ super-droplets with $\xi = 1$ to the cell. Their radii are selected by randomly drawing $\ln(r)$ from the distribution $C n(\ln(r)) H(\ln(r) - \ln(r_{\max}))$, where $C$ is a normalizing constant and $H(x)$ is the Heaviside step function. This makes the actual number of SDs $N$ higher than the prescribed value $N_{SD}$, typically by ca. 1%. We do not add SDs from the small tail of the distribution, because very small droplets are of little importance for rain formation. In this type of simulations, the timestep length is constant $\Delta t = 1$ s. It is not adapted, as it is done in the "one-to-one" simulations, to make the simulation computationally more efficient. Using constant timestep length can make the coalescence probability exceed unity. If it does, it is assumed that a pair of SDs collides more than once during the timestep (Shima et al., 2009). Then, the procedure for calculating post-collisional parameters (Eq. 1) is applied $\tilde{\gamma} = \min(\gamma, \xi_i/\xi_j)$ times, where $\gamma \geq 1$ is the number of collisions between the $i$-th and the $j$-th SD and $\xi_i \geq \xi_j$. Such handling of multiple collisions can cause the expected number of collisions to be lower than the real one if $\gamma > \xi_i/\xi_j$. Another inconsistency is that rigorously, the probability of collision between SDs should change after each of the $\tilde{\gamma}$ collisions. For these reasons timestep length should be carefully selected so that multiple collisions do not happen often. If two SDs with identical $\xi$ collide, multiplicity of one of them drops to zero. Then, the SD with $\xi = 0$ is used to split the SD with the largest $\xi$ in the system into two. This is slightly different than in the Shima et al. (2009) algorithm, in which a SD with $\xi = 0$ is used to split the other SD that came out of the collision that caused the multiplicity to drop to zero. Super-droplets are discarded after collision only if all other SDs have $\xi = 1$. We use an implementation of the SDM from the libcloudph++ library (Arabas et al., 2015). It is an open-source project available at https://github.com/igfuw/libcloudphxx.

## 3 Comparison of the "one-to-one" SDM with the master equation

The goal of this Section is to show that the "one-to-one" SDM is at the same level of precision as the master equation. To this end, we calculate average droplet size distribution and standard deviation of mass of the largest droplet from an ensemble of "one-to-one" simulations. As a reference, we use the results from Alfonso and Raga (2017), who used the master equation approach to study the sol-gel transition. In a system of aggregating particles, the sol-gel transition (gelation) occurs when most of the total mass is located in a single agglomerate (Leyvraz, 2003). For some forms of the coalescence kernel, the Smoluchowski equation is known not to conserve mass after the transition. Alfonso and Raga (2017) present numerical solutions of the master equation for a small cloud volume undergoing the sol-gel transition, for which the Smoluchowski equation is not valid. We perform simulations for the same setup to test if the "one-to-one" simulations are in agreement with the master equation approach. Consider a 1 cm$^3$ volume containing 20 droplets with the radius of 17 μm and 10 droplets of radius 21.4 μm. Gravitational collision kernel is used with collision efficiencies from Hall (1980). Collision efficiencies are bilinearly interpolated in the radius - ratio of radii space. Droplet terminal velocities are calculated using the formula from Beard (1976).

Figure 1 shows the average mass distribution obtained using the "one-to-one" simulations with and without linear sampling of collision pairs. The average is calculated from an ensemble of $\Omega = 10^4$ realizations for each case. In simulations without linear sampling, all $N(N-1)/2$ collision pairs are considered and a constant timestep $\Delta t = 0.1$ s is used. Both approaches give similar results, what shows that the linear sampling technique does not affect the average number of collisions. In addition,

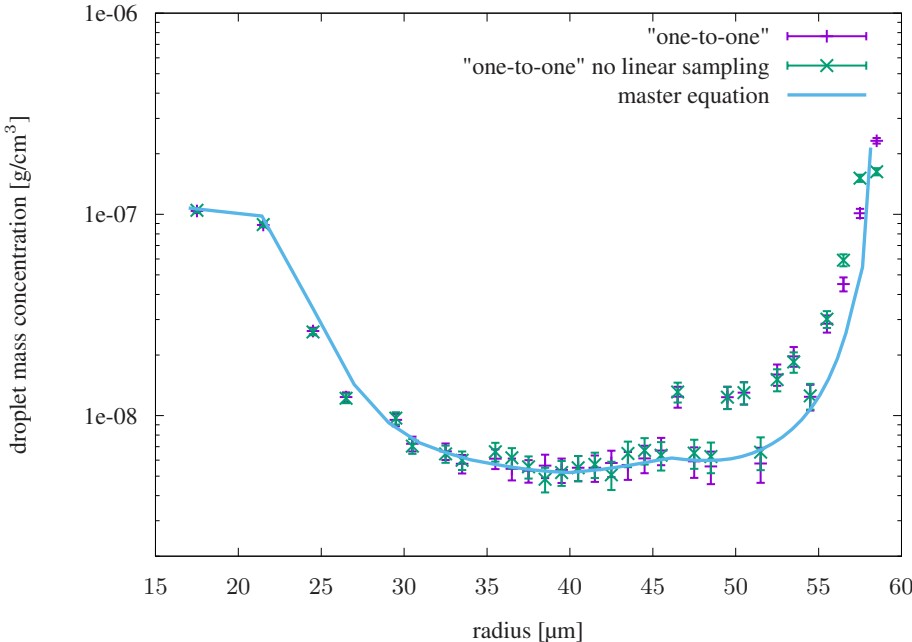

**Figure 1.** Mass of droplets per size bin at $t = 2500$ s. Bins are 1 μm wide. Points depict an averaged result of $\Omega = 10^4$ "one-to-one" simulations with and without linear sampling of collision pairs. Error bars show a 95% confidence interval. Line depicts a numerical solution of the master equation (see Fig. 8 in Alfonso and Raga (2017), data courtesy of L. Alfonso).

the "one-to-one" simulations are compared with the master equation solved using the method of Alfonso and Raga (2017). Both approaches are generally in agreement, with some differences at the large end of the distribution. These differences may be caused by the way how the coalescence efficiency tables are interpolated. Another possible source of discrepancies is the numerical diffusion present in the finite-differences method of Alfonso (2015). To test if the "one-to-one" method also gives

5   correct fluctuations in the number of collisions, relative standard deviation of mass of the largest droplet $\sigma(m_{\mathrm{max}})/\langle m_{\mathrm{max}}\rangle$ is plotted in Fig. 2. This value is of interest, because the sol-gel transition time coincides with the time at which $\sigma(m_{\mathrm{max}})/\langle m_{\mathrm{max}}\rangle$ reaches maximum (Leyvraz, 2003; Alfonso and Raga, 2017). In Fig. 2, "one-to-one" simulations, with and without linear sampling, are compared with results of the master equation approach presented in Alfonso and Raga (2017). Please note that Alfonso and Raga (2017) obtained values of $\sigma(m_{\mathrm{max}})/\langle m_{\mathrm{max}}\rangle$ from an ensemble of SSA simulations rather than by solving

10   the master equation, as was the case in Fig. 1. As in Fig. 1, we do not observe any negative effect of using the linear sampling technique and the "one-to-one" simulations compare relatively well with the SSA. Possible sources of discrepancies are the same as in Fig. 1. Judging from Figs. 1 and 2, we conclude that the "one-to-one" approach is in agreement with the master equation approach. It accounts for the correlations in the number of droplets per size-bin and as such is more fundamental than the Smoluchowski equation approach.

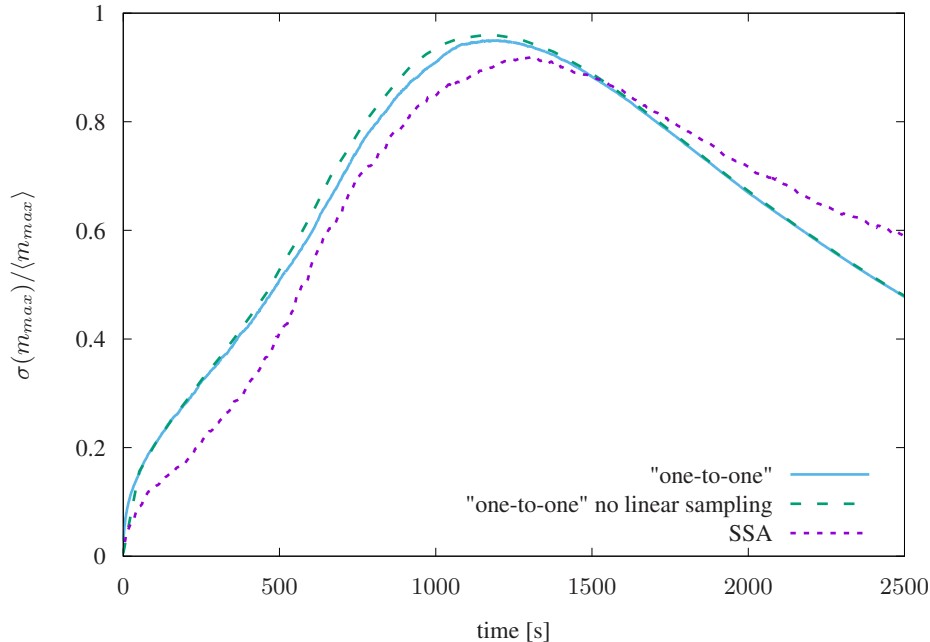

**Figure 2.** Relative standard deviation of mass of the largest droplet in the system. Details of the SDM simulations are given in the caption of Fig. 1. Size of the ensemble of SSA simulations is $\Omega = 10^3$. The SSA results are taken from Fig.7 in Alfonso and Raga (2017) (data courtesy of L. Alfonso).

The "one-to-one" SDM with linear sampling is computationally more efficient than solving the master equation directly, or using the SSA. It also puts no constraints on the initial distribution of droplets. Therefore we can use SDM to predict gelation times for larger systems and more realistic initial conditions. We use an initial droplet distribution that is exponential in mass $n(m) = \frac{n_0}{\overline{m}}exp(-m/\overline{m})$, where $n(m)dm$ is the number of droplets in mass range $(m, m+dm)$ in unit volume, $n_0 = 142$

5  cm$^{-3}$ and $\overline{m}$ is the mass of a droplet with radius $\overline{r} = 15$ µm. This is the same distribution as in Onishi et al. (2015). The total initial number of droplets in the system is $N_0 = n_0 V$. Results of the "one-to-one" simulations for $N_0$ up to $10^5$ are shown in Fig. 3. For $N_0 \geq 10^2$, the relative standard deviation of mass of the largest droplet decreases with increasing cell size. This can be understood if we look at a larger cell as an ensemble of ten smaller cells. Comparing between independent realizations, variability in the size of the single, largest droplet will be smaller if this droplet is selected from ten cells in each realization than if it was selected from only a single cell per realization. Interestingly, for $N_0 = 10^5$ an inflection point appears around

10  $t = 500$ s. It is not seen in smaller cells. This indicates that some new source of variability is introduced. We believe that it is associated with collisions between large rain drops. We will come back to this in Sec. 5. Intuitively, we would expect the time for most of the mass to accumulate in a single agglomerate to increase with increasing cell size. This turns out to be true for cells with $N_0 > 10^3$. For cell sizes $10^2 < N_0 < 10^3$ gelation time is approximately the same, around 300 s.

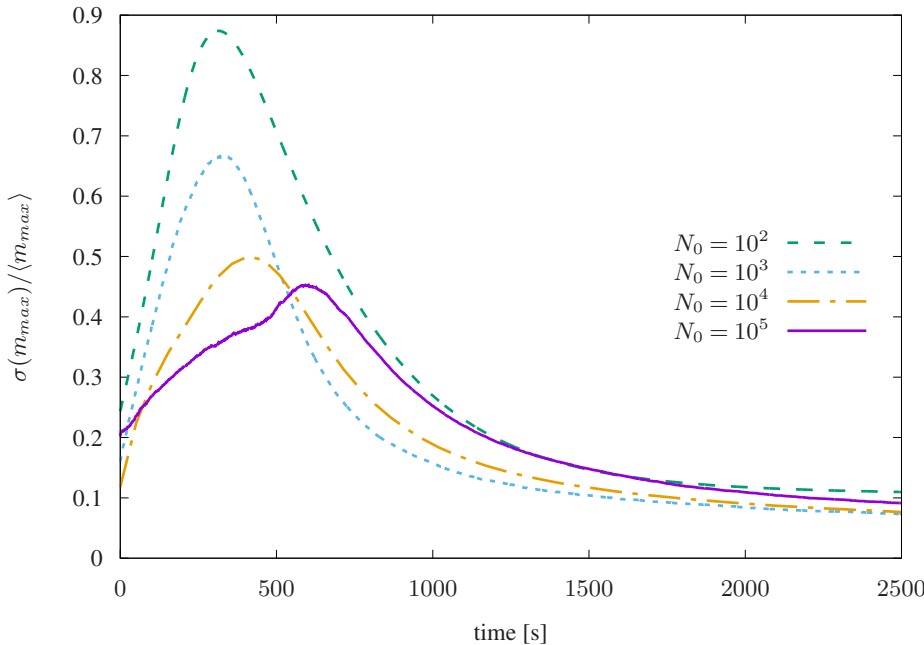

**Figure 3.** Relative standard deviation of mass of the largest droplet for different cell sizes. Estimated from ensembles of $\Omega = 10^4$ "one-to-one" simulations for each value of $N_0$.

## 4  Fluctuations in conversion to rain drops and validity of the "constant SD" SDM

Fluctuations in time of conversion of cloud droplets to rain drops were studied using direct numerical simulations by Onishi et al. (2015). Following their notation, by $t_{10\%}$ we denote time after which $10\%$ of mass of cloud droplets is turned into droplets with $r > 40$ μm. Droplets of this size should then quickly grow through coalescence. The time $t_{10\%}$ is used as a measure of efficiency of rain production. We will compare results of the "one-to-one" simulations with DNS and try to determine how many SDs are needed in the "constant SD" simulations to accurately represent coalescence. The initial droplet distribution and coalescence kernel are the same as in Sec. 3.

In Fig. 4, values of mean $t_{10\%}$ for different initial number of droplets are plotted against the number of SDs. Results of both the "one-to-one" (rightmost points in each series) and the "constant SD" (rest of the points in the series) simulations are presented. For comparison, $t_{10\%}$ obtained by solving the Smoluchowski equation using the Bott algorithm is plotted (Bott, 1998). In the Bott algorithm, we used $\Delta t = 1$ s and mass bin spacing $m_{i+1} = 2^{1/10} m_i$. The same parameters were used in any Bott simulation presented in this manuscript. Convergence tests were done for each case. The "one-to-one" results converge with increasing cell volume (i.e. increasing $N_0$) to a value higher than the Smoluchowski result. The difference is probably caused by the numerical diffusion of the Bott algorithm. In the "constant SD" simulations, error caused by using SDs with $\xi > 1$ weakly depends on the cell size. Using $10^3$ SDs gives $\langle t_{10\%} \rangle$ within $1\%$ of the "one-to-one" value. Using $10^2$ SDs

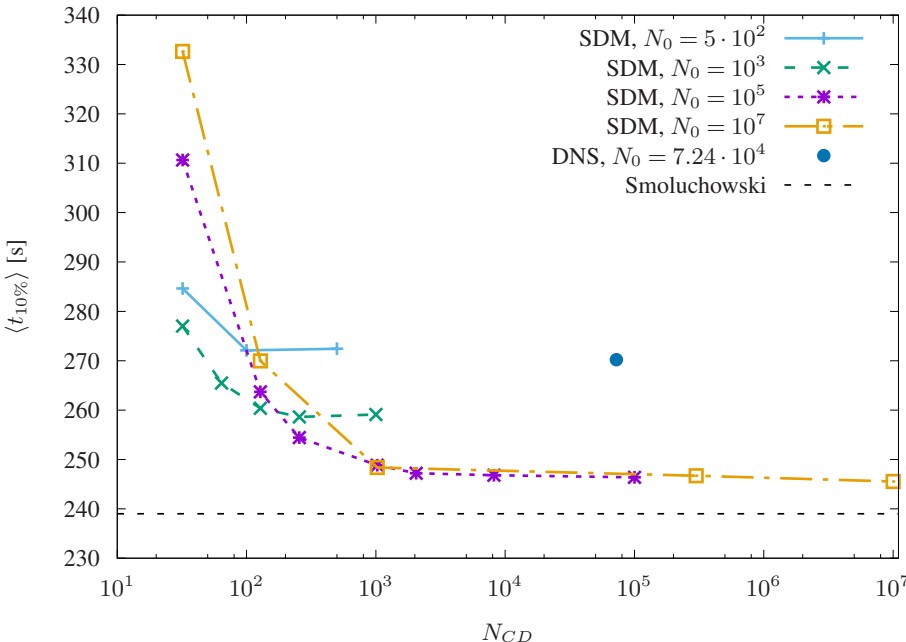

**Figure 4.** Mean $t_{10\%}$ for different cell sizes and different numbers of computational droplets $N_{CD}$. In SDM simulations, $N_{CD} = N_{SD}$ and in DNS, $N_{CD} = N_0$. The single DNS result is taken from Onishi et al. (2015) (the NoT-HI case therein). Ensemble sizes are $\Omega \geq 10^3$ for SDM simulations and $\Omega = 10^2$ for DNS. The 95% confidence intervals are smaller than plotted points. The rightmost point in each SDM series comes from the "one-to-one" simulations. Other points in SDM series come from the "constant SD" simulations with various values of $N_{SD}$. The horizontal line is a value obtained by numerically solving the Smoluchowski equation using the flux method from Bott (1998).

causes about 10% difference. This shows that, in terms of computational cost, it is relatively cheap to obtain a good estimate of the average result of coalescence using the SDM. The SDM results are also compared with the results of DNS, in which air turbulence was not modelled, but hydrodynamic interactions between droplets were accounted for. We choose this kind of DNS, because it should be well described by the Hall kernel that is used in the SDM and in the Smoluchowski equation. It

5  turns out that the Hall kernel gives too short autoconversion times. The same issue was observed by Onishi et al. (2015) (cf. Fig. 1(b) therein).

To analyze the amplification of fluctuations in the "constant SD" method, we plot the relative standard deviation of $t_{10\%}$ in Fig. 5. For reference, results of DNS from Onishi et al. (2015) are shown. Results from our "one-to-one" simulations are in good agreement with the DNS. Small discrepancies are probably caused by the fact that the DNS included turbulence of

10  various strength for different $N_0$. Results of the "one-to-one" simulations were fitted with the function $\alpha\sqrt{1/N_0}$ , resulting in $\alpha = 6$. Figure 5 also presents fluctuations in the "constant SD" simulations for various $N_{SD}$. This type of simulations gives correct amplitude of fluctuations only for relatively low values of the ratio $N_0/N_{SD}$. For constant $N_{SD}$, as $N_0$ increases, the amplitude of fluctuations decreases correctly. Then, above some critical value of the $N_0/N_{SD}$ ratio, fluctuations stop to

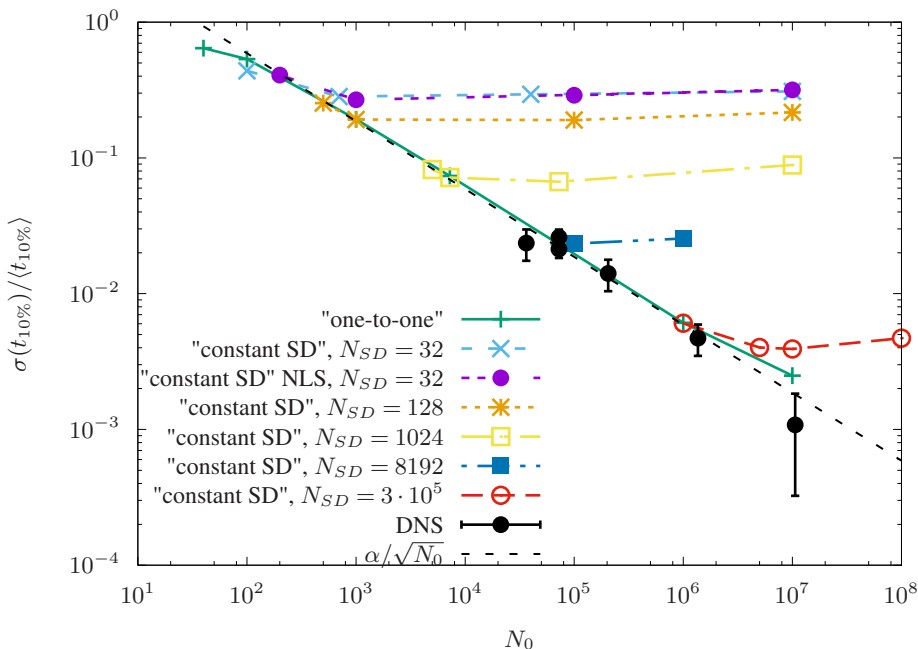

**Figure 5.** Relative standard deviation of $t_{10\%}$ against cell size. SDM results are based on samples of at least $10^3$ realizations. DNS results are taken from Onishi et al. (2015). Where not shown, errorbars are smaller than plotted points. The value $\alpha = 6$ was obtained by curve fitting to the "one-to-one" results. The acronym NLS in the legend stands for "no linear sampling".

decrease and remain constant independent of the cell size. This is a result of introducing unrealistic correlations between droplet sizes, which is a consequence of using low number of computational particles (Bayewitz et al., 1974). To show that the linear sampling technique does not contribute to this effect, we plot result of a "constant SD" simulation without linear sampling for $N_{SD} = 32$, which is the same as for $N_{SD} = 32$ with linear sampling. We show the limiting, minimal value of relative standard

5   deviation of $t_{10\%}$ in Fig. 6. It decreases as $\beta\sqrt{1/N_{SD}}$, with $\beta = 2$. By comparing it with $\alpha = 6$, we conclude that in order to obtain correct fluctuations in $t_{10\%}$ using "constant SD" simulations, the number of SDs has to be $N_{SD} \geq \frac{1}{9}N_0$. Using so many SDs is not feasible in Large Eddy Simulations (LES), but is possible in smaller scale simulations. Also, knowing $\alpha$ and $\beta$, we can estimate the magnitude of fluctuation amplification in the "constant SD" SDM.

## 5   Validity of the Smoluchowski equation

10   The Smoluchowski equation presents a mean-field description of the evolution of the size spectrum. It is exact only in the thermodynamic limit ($V \rightarrow \infty$). We will try to determine minimal cell size for which the Smoluchowski equation can be used without introducing major errors. To do so, we analyze the evolution of $\theta$, the ratio of rain water ($r \geq 40$ μm) content to the

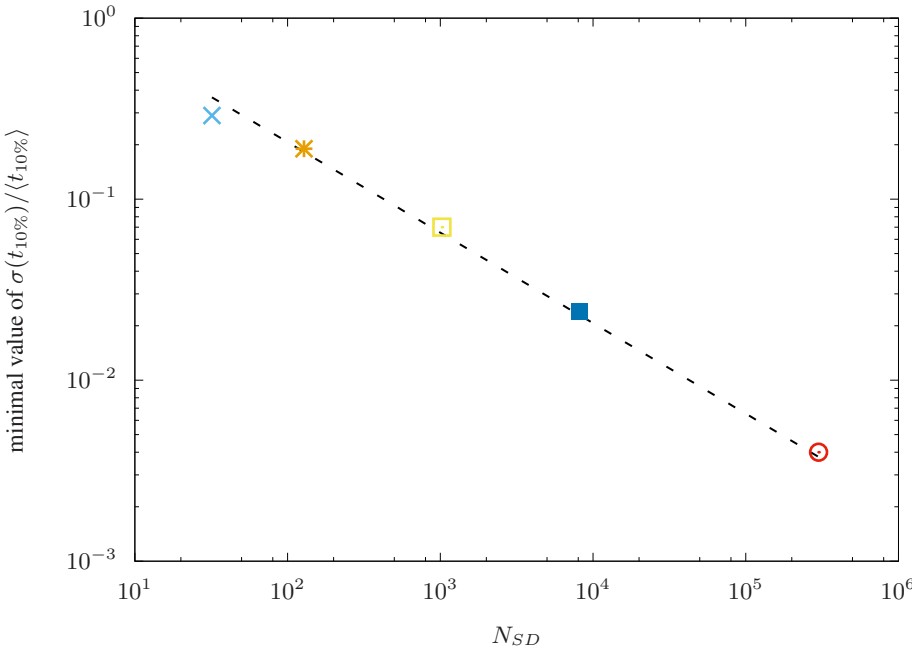

**Figure 6.** Points depict the minimal, limiting value of the relative standard deviation of $t_{10\%}$ for a given number of super-droplets in "constant SD" simulations. For each value of $N_{SD}$, the minimal value of $\sigma(t_{10\%})/\langle t_{10\%}\rangle$ is calculated as an average of the respective points to the right of the $\alpha/\sqrt{N_0}$ curve in Fig. 5. Line depicts the fitted function $\beta/\sqrt{N_{SD}}$ with $\beta = 2$.

total water content. Onishi et al. (2015) denote this value by $\tau$. We do not adopt this notation to avoid confusion with the characteristic time.

We compare results of the "one-to-one" simulations with solutions of the Smoluchowski equation for two cases - with fast and with slow rain development. In both cases collision efficiencies for large droplets are taken from Hall (1980) and for small

droplets from Davis (1972). In simulations with fast development of rain, we use the same initial distribution as in Secs. 3 and 4. As seen in Fig. 7, the Smoluchowski equation gives correct mean rain development rate for cells with $N_0 \geq 10^4$. The Smoluchowski curve is slightly shifted left, probably due to the numerical diffusion of the Bott algorithm, as discussed in Sec. 4. In cells smaller than $N_0 = 10^4$, rain develops slower than predicted by the Smoluchowski equation. Agreement of stochastic coalescence in large cells with the Smoluchowski equation for a similar initial distribution was shown using the SSA by

Seesselberg et al. (1996). Onishi et al. (2015) present figures similar to Fig. 7, but obtained from DNS runs for $N_0 = 7.24 \cdot 10^4$ (Fig. 1(b) therein). They show good agreement between DNS and the Smoluchowski equation with the kernel of Long (1974), at least up to $t = 330$ s. If the Hall kernel is used in the Smoluchowski equation, autoconversion is quicker than in the DNS, as discussed in Sec. 4.

Next, we validate the Smoluchowski equation in a setup with slow rain development. This time the initial droplet size

distribution is below the size gap, i.e. the range of radii for which both collisional and condensational growths are slow. We use

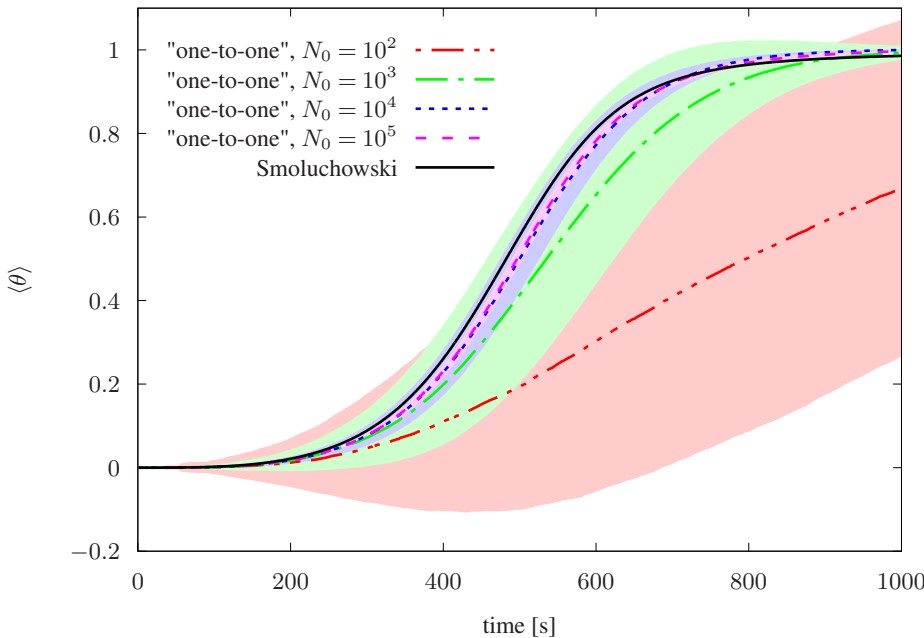

**Figure 7.** Rain content ratio $\theta$ for different cell sizes averaged over ensembles of $\Omega = 10^3$ simulations. Shaded regions show one standard deviation interval.

$\bar{r} = 9.3$ μm and $n_0 = 297$ cm$^{-3}$ as in Wang et al. (2006). In addition, we cut the distribution to 0 for $r \geq 20$ μm. This cutoff is used in the SDM modelling as well as when solving the Smoluchowski equation. That way we get rid of the occasional very large SDs present at $t = 0$ in some realizations of the SDM. For these initial conditions, rain development takes much longer and fluctuations can play a bigger role. Results are presented in Fig. 8. Again, we see convergence of the "one-to-one"

simulations to the Smoluchowski result, but in this case the cell has to be larger ($N_0 \geq 10^7$) for the Smoluchowski equation to be valid. The way how the "one-to-one" curves converge to the Smoluchowski curve is interesting. As in the first case, in smaller cells rain appears later than in larger cells. On the other hand, the rain formation rate (the slope of the curves in Fig. 8) in smaller cells starts to decrease at higher values of $\theta$ than in larger cells. In consequence, smaller cells can produce higher rain ratio than larger ones, although they started producing rain later (e.g. compare curves for $N_0 = 10^5$ and $N_0 = 10^7$

for $t > 4200$ s). The decrease in the rain formation rate is associated with the decrease in the concentration of rain drops $n_r$, plotted in Fig. 9. Number of rain drops decreases due to collisions between drops from this category. A single drop that is produced in such collision is less efficient at scavenging cloud droplets than the two pre-collision drops. In result, growth rate of $\theta$ decreases. Using large well-mixed volumes may introduce additional, unrealistic rain-rain collisions. Consider two droplets within a large cell that independently grow to the rain category. They have to be separated enough not to deplete liquid

water from each other's surrounding as they grow. If we assume that the cell is well-mixed, they can collide immediately after becoming rain drops and generate an even larger drop. In reality, they could collide only after some time after becoming rain

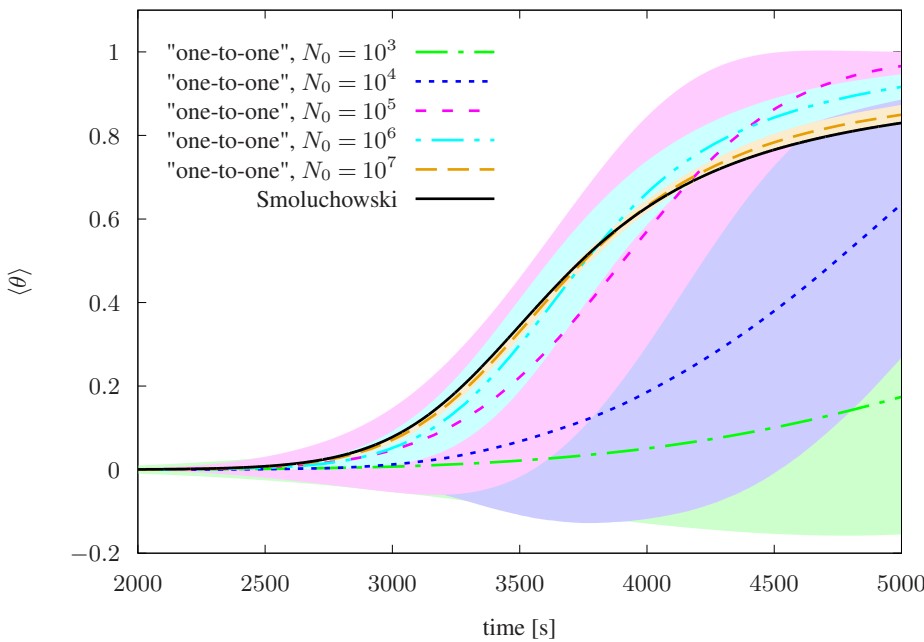

**Figure 8.** As in Fig. 7, but for an initial distribution with $\bar{r} = 9.3$ μm, $n_0 = 297$ cm$^{-3}$ and a cutoff at $r = 20$ μm. The ensemble size is $\Omega = 10^8/N_0$.

drops, because first they would need to overcome the initial separation. This means that using large well-mixed volumes, e.g. in the Smoluchowski equation, may result in an artificial decrease in the number of rain drops and an underestimation of mass of rain produced.

In coalescence cells with $N_0 \leq 10^4$, we do not observe the decrease in the number of rain drops within 5000 s, probably
5 because sizes of rain drops are similar. In larger cells, more rain drops with a broader distribution are formed. In consequence, they collide more often, which decreases their number and the rate of collection of cloud droplets. It is likely that the additional rain-rain collisions in large volumes are responsible for the additional inflection point around $t = 500$ s in the plot of the relative standard deviation of the largest droplet mass for $N_0 = 10^5$ (cf. Fig. 3). They could also lead to the deviation from the $\sim 1/\sqrt{N_0}$ scaling shown in Fig. 5. Fluctuations in cells with $N_0 = 10^7$ are greater than predicted using this scaling. We also
10 observe that for $t \leq 3000$ s, the number of rain drops does not depend on cell size (c.f. Fig. 9), but the mass of rain water does (c.f. Fig. 8). In larger cells rain drops acquire larger sizes through collisions with cloud droplets, but rate of autoconversion of cloud droplets to rain drops is not affected much by cell size.

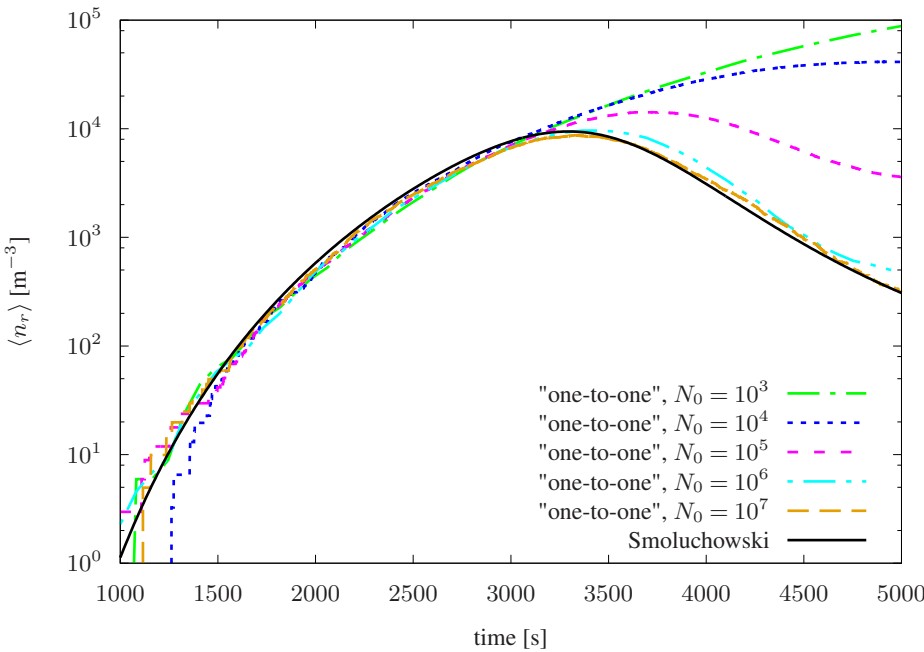

**Figure 9.** Mean concentration of rain drops from the same simulations as in Fig. 8.

**Table 1.** Average and standard deviation of time (in seconds) for lucky realizations to produce a single drop with $r \geq 40$ μm. $\gamma$ is fraction of the fastest realizations out of all $\Omega$ realizations that were used to compute the respective $\langle t_{40} \rangle_\gamma$ and $\sigma(t_{40})_\gamma$. The sub-ensemble size $\gamma\Omega$ is shown to give an idea about precision of estimation of the respective statistics.

| $N_0$ | $\gamma = 10^{-4}$ | | | $\gamma = 10^{-3}$ | | | $\gamma = 10^{-2}$ | | | $\gamma = 10^{-1}$ | | | $\gamma = 1$ | | |
|---|---|---|---|---|---|---|---|---|---|---|---|---|---|---|---|
| | $\langle t_{40} \rangle_\gamma$ | $\sigma(t_{40})_\gamma$ | $\gamma\Omega$ | $\langle t_{40} \rangle_\gamma$ | $\sigma(t_{40})_\gamma$ | $\gamma\Omega$ | $\langle t_{40} \rangle_\gamma$ | $\sigma(t_{40})_\gamma$ | $\gamma\Omega$ | $\langle t_{40} \rangle_\gamma$ | $\sigma(t_{40})_\gamma$ | $\gamma\Omega$ | $\langle t_{40} \rangle_\gamma$ | $\sigma(t_{40})_\gamma$ | $\gamma\Omega$ |
| $10^2$ | 2052 | 212 | 10 | 2930 | 356 | 10 | 4053 | 517 | $10^2$ | 6365 | 1158 | $10^3$ | 14777 | 6099 | $10^3$ |
| $10^3$ | 1366 | 120 | $10^2$ | 1762 | 170 | $10^3$ | 2400 | 267 | $10^4$ | 3440 | 505 | $10^5$ | 6500 | 1700 | $10^6$ |
| $10^4$ | 1089 | 173 | 3 | 1336 | 103 | 10 | 1717 | 176 | $10^2$ | 2354 | 276 | $10^3$ | 3912 | 764 | $10^4$ |
| $10^5$ | 946 | 33 | 2 | 1090 | 60 | 20 | 1334 | 85 | 200 | 1721 | 169 | 2000 | 2552 | 415 | $10^4$ |
| $10^6$ | | | | | | | 1038 | 165 | 2 | 1301 | 176 | 20 | 1831 | 277 | $10^2$ |

## 6 Lucky droplets

There is a well-established idea that some droplets undergo series of unlikely collisions and grow much faster than an average droplet (Telford, 1955; Scott, 1967; Marcus, 1968; Robertson, 1974; Mason, 2010). These few lucky droplets are argued to be responsible for droplet spectra broadening and rain forming quicker than predicted by the Smoluchowski equation. Luck is supposed to be especially important during crossing of the size gap, when collisions happen rarely (Robertson, 1974; Kostinski and Shaw, 2005). A single droplet that would cross the size gap through lucky collisions could then initiate a cascade

of collisions. Theoretical estimation of the "luck factor" was presented in Kostinski and Shaw (2005). We use the "one-to-one" simulations to test predictions from that paper.

We are interested in time $t_{40}$ it takes until the first droplet grows to $r = 40$ μm. We perform ensembles of simulations for different cell sizes $N_0$. Size of an ensemble is denoted by $\Omega$. The initial distribution is the same as in the second case in Sec. 5. The mean radius is $\bar{r} = 9.3$ μm, well below the size gap. The liquid water content is $1\mathrm{g/cm}^3$ and the concentration is 297 cm$^{-3}$, so the smallest cell that has enough water to produce a droplet with $r = 40$ μm is $N_0 \approx 80$. Therefore the smallest cell size we consider is $N_0 = 10^2$. For each value of $N_0$, we select sub-ensembles of luckiest realizations, i.e. those with the smallest $t_{40}$. We consider sub-ensembles of size $\gamma\Omega$ with $log_{10}(\gamma) = -4, -3, -2, -1, 0$. In each sub-ensemble, we calculate the mean $\langle t_{40} \rangle_\gamma$ and the standard deviation $\sigma(t_{40})_\gamma$, where the subscript $\gamma$ denotes the size of the sub-ensemble from which the statistics were calculated. The results for different cell sizes are shown in Tab. 1.

There is a large variability in $\langle t_{40} \rangle_\gamma$ with cell size. This is caused by the fact that $t_{40}$ depends only on a single largest droplet. Larger cells contain more droplets, so probability of producing single large droplet increases with cell size. We notice that $\langle t_{40} \rangle_\gamma$ is approximately the same along the diagonals of Tab. 1. For example, a cell containing $10^6$ droplets on average will produce first rain droplet in 30 minutes. If we divided it into 10 cells with $10^5$ droplets each, the luckiest one out of ten would also produce a droplet in 30 minutes on average. This shows that using large coalescence cells does not affect formation of first rain drops. The differences discussed in previous Sections emerge later, when there are already some rain drops that can collide with each other. To depict that size of a coalescence cell does not affect formation of first rain drops, it is helpful to think about the test presented in this Section as a simulation of a large system initially containing $N_{\mathrm{tot}} = N_0/\gamma$ droplets divided into coalescence cells of size $N_0$. We are interested in the mean time $\langle t_{40} \rangle$ for the first droplet out of all $N_{\mathrm{tot}}$ droplets to grow to $r = 40$ μm. In Fig. 10 we plot $\langle t_{40} \rangle$ against $N_{\mathrm{tot}}$ for different sizes of coalescence cell. It is seen that $\langle t_{40} \rangle$ does not depend on $N_0$. Even using a coalescence cell with $N_0 = 10^2$, in which there is barely enough water to produce a drop with $r = 40$ μm, does not change the results.

Next we calculate the "luck factor", i.e. how much faster the luckiest droplets grow to $r = 40$ μm compared to the average droplets. To calculate it we use the data for $N_0 = 10^2$, because cells of this size can produce only a single droplet with $r = 40$ μm. Larger cells behave like an ensemble of cells with $N_0 = 10^2$ as far as $t_{40}$ is concerned, so calculating the "luck factor" using $t_{40}$ from larger cells would tell us how much faster the luckiest ensemble of droplets produces a rain droplet, compared to an average ensemble - a quantity that we are not interested in. We find $\langle t_{40} \rangle_1 / \langle t_{40} \rangle_{10^{-3}} \approx 5$ and $\langle t_{40} \rangle_1 / \langle t_{40} \rangle_{10^{-5}} \approx 11$. The value of $\langle t_{40} \rangle_{10^{-5}}$ was estimated at 1366 s based on values along the diagonal for larger $\gamma$ and larger $N_0$. Kostinski and Shaw (2005) estimate that the luckiest $10^{-3}$ fraction of droplets should cross the size gap around six times faster than average, while the luckiest $10^{-5}$ around nine times faster. These values are in good agreement with our observations.

## 7   Size of a well-mixed coalescence cell

In the previous Sections we have seen that size of the coalescence cell has a profound impact on the evolution of the system. In this Section we estimate the size of a cell that can be assumed to be well-mixed. All methods in which probability of collision

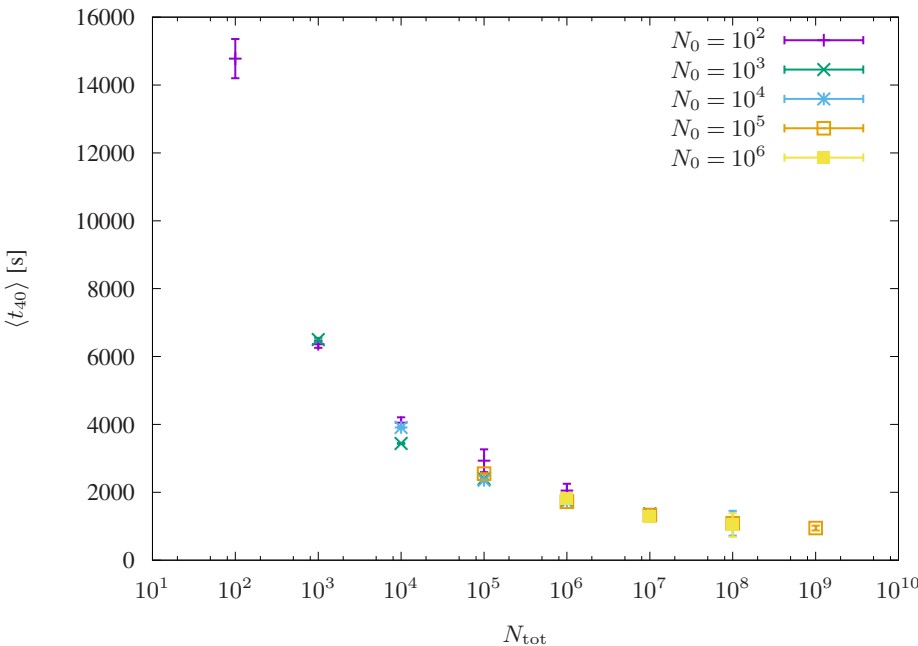

**Figure 10.** Mean time until a system of $N_{tot}$ droplets produces first droplet with $r = 40\mu$m. The system is divided into coalescence cells of size $N_0$. The figure is based on the results of "one-to-one" simulations given in Tab. 1.

of droplets depends only on the instantaneous state of the cell and not on its history rely on the assumption that the cell is well-mixed. This includes the master equation, SSA, SDM as well as the Smoluchowski equation. The assumption that a cell is well-mixed is valid if $\tau_{mix} \ll \tau_{coal}$, where $\tau_{coal}$ and $\tau_{mix}$ are the characteristic times for coalescence and cell homogenization, respectively (Lehmann et al., 2009; Gillespie et al., 2014). By well-mixed we mean that droplets should be distributed homoge-
neously within the cell before every collision. Droplet coalescence generates inhomogeneities, i.e. correlations between droplet positions and sizes.

Rigorously, characteristic time for coalescence is the mean time between coalescence events, as in diffusion-limited chemical systems (Gillespie et al., 2014). To estimate its magnitude, consider a single large collector droplet falling through a field of smaller droplets. Using geometric coalescence kernel with efficiency $E$, the mean time between collisions is
$\tau_{coal} = (E\pi(r_l + r_s)^2 v_r n_s)^{-1}$, where $r_l$ and $r_s$ are radii of large and small droplets, $v_r$ is the relative velocity and $n_s$ is the concentration of small droplets. For $r_l = 100$ µm, $r_s = 10$ µm, $v_r = 70$ cm/s, $E = 1$ and $n_s = 100$ cm$^{-3}$ we get $\tau_{coal} \approx 0.4$ s.

Droplets in the cell can be mixed through turbulence. Turbulence acts similarly to diffusion and its characteristic time for mixing is $\tau_{mix}^t = (V^{(2/3)}/\varepsilon)^{(1/3)}$, where $V$ is cell volume and $\varepsilon$ is turbulent energy dissipation rate (Lehmann et al., 2009). Turbulent energy dissipation rate in clouds is in the range from 10 cm$^2$/s$^3$ for stratocumulus clouds to $10^3$ cm$^2$/s$^3$
for cumulonimbus clouds (Malinowski et al., 2013; Grabowski and Wang, 2013). Let us assume that $\tau_{mix}^t \ll \tau_{coal}$ is satisfied if $\tau_{mix}^t = 0.1\tau_{coal}$. Even in the most turbulent clouds, this means that the coalescence cell has to be very small $V \approx 1.5 \cdot 10^{-2}$cm$^3$.

On average, this volume would contain around one droplet, depending on concentration of droplets. For such small coalescence volumes, the Smoluchowski is not valid and SDM would be very cumbersome, because extremely short timesteps would be required. To use larger cells, we need to choose some less strict value of characteristic time of coalescence. Some larger cell size, that would be approximately well-mixed, could be found phenomenologically through fine-grid simulations including droplet motion. One example of such reference simulations are DNS runs from Onishi et al. (2015) discussed in Sec. 5. They prove that in the case with $\bar{r} = 15$ µm, the Smoluchowski equation gives correct results. This suggests that cells with $N_0 \geq 10^4$ can be used in this case.

Another process that can mix droplets is sedimentation. It is difficult to assess its timescale, because it strongly depends on droplet sizes. Droplets of similar sizes are not mixed by sedimentation, but it is efficient at mixing rain drops with cloud droplets. We can expect that it would prevent depletion of cloud droplets in the surrounding of a rain droplet that was observed for smallest cells in Secs. 3 and 6. Sedimentation acts only in one direction, so it could only allow us to use cells larger only in the vertical direction.

## 8   Conclusions

The super-droplet method can exactly represent stochastic coalescence in a well-mixed volume. It was compared with the master equation approach (see Sec. 3) and with direct numerical simulations (see Sec. 4). Precision of the SDM is controlled by the number of super-droplets used. Fluctuations in the autoconversion time are represented well if $N_{SD} \geq N_0/9$. Using smaller $N_{SD}$ increases standard deviation of autoconversion time by a factor $\frac{1}{3}\sqrt{N_0/N_{SD}}$ (cf. Sec. 4). It is computationally less expensive to correctly reproduce mean autoconversion time. Using $N_{SD} = 10^3$ gives mean results within a 1% margin, while using $N_{SD} = 10^2$ - within 10%.

The SDM was used to study stochastic coalescence for two initial droplet size distributions - with small ($\bar{r} = 9.3$ µm) and with large ($\bar{r} = 15$ µm) droplets. They result in slow and fast rain formation, respectively. Dependence of the system behavior on size of the well-mixed coalescence cell was observed, especially in the small droplets case. Cell size not only affects fluctuations in the observables, but also their expected values. If the coalescence cell is small, sizes of droplets are strongly correlated and depletion of cloud water plays an important role. In real clouds, these two effects are probably not manifested, because collector drop sedimentation acts against them. In relatively large cells, rain drops collide with each other more often than in small cells. This leads to a reduction in the rate of conversion of cloud water to rain water, because scavenging of cloud droplets becomes less efficient. In consequence, highest rain content is produced in cells of intermediate sizes. Possibly, these additional rain-rain collisions can be justified by turbulent droplet motion and sedimentation, but they also might be an artefact caused by using an unrealistically large well-mixed volume. Fine-grid computer modeling with explicit droplet motion could be used to resolve this issue. If the additional collisions were found to be unrealistic, it would mean that cloud models that use large well-mixed cells, e.g. by using the Smoluchowski equation, produce too little rain.

The additional rain-rain collisions do not affect results if droplets are initially large. Then, collisions of cloud and rain drops and between cloud droplets are frequent, so the increase in the rate of collisions between rain drops is not important. The

mean behavior of the system converges to the Smoluchowski equation results with increasing cell size. Good agreement with it is found for cells with $N_0 \geq 10^4$. The picture is different if droplets are initially small. Conversion of cloud droplets into rain drops is slow, so the decrease in rain drop concentration due to the additional collisions is relatively more important. The Smoluchowski equation is found to be valid for $N_0 \geq 10^7$ for the slow-coalescence case. One could expect that condensational

growth leads to initial conditions with high radii of droplets, for which the additional collisions are not important. Li et al. (2017) have shown that condensation can regulate differences between Eulerian and Lagrangian coalescence schemes. Discrepancies between these schemes that they observed in simulations with condensation and coalescence were smaller than in pure coalescence simulations.

Another aspect of the slow-coalescence scenario is that in it, some lucky droplets can grow much faster than average droplets.

We found that a single luckiest droplet out of a thousand grows 5 times faster than average and the luckiest out of a hundred thousand - 11 times faster. These values are are in good agreement with the analytical estimation of Kostinski and Shaw (2005).

We estimate a well-mixed (with respect to coalescence) volume in the most turbulent clouds to be only $1.5 \cdot 10^{-2}$ cm$^3$. It is of the order of the volume occupied by a single droplet. Larger cells can be assumed to be only approximately well-mixed. For example, in the fast-coalescence case, DNS modeling gives the same results as the Smoluchowski equation (Onishi et al.,

2015). Box model simulations using well-mixed volume with $N_0 = 10^4$ droplets also gives the same results. Therefore it can be assumed that such volume is approximately well-mixed in the case of fast coalescence. In the slow-coalescence case, the well-mixed volume needs to be larger than in the fast-coalescence case for the Smoluchowski equation to be valid. Size of an approximately well-mixed cell for this case can be determined using DNS with initially small droplets. Volume of cells used in LES is typically ten orders of magnitude larger than a well-mixed volume. The LES cells do not necessarily have to be

well-mixed. It is sufficient if they are homogeneous, i.e. they are an ensemble of identical, approximately well-mixed sub-cells. Some statistical moments for such ensembles were presented in this work. In general, it is not clear what could be the size of these sub-cells and if the Smoluchowski equation is valid for them.

## 9 Code availability

Simulation code is available at https://github.com/pdziekan/coal_fluctu. The libcloudph++ library is available at

25 https://github.com/igfuw/libcloudphxx.

*Acknowledgements.* This study was financed from Poland's National Science Center "POLONEZ 1" grant 2015/19/P/ST10/02596 (this project has received funding from the European Union's Horizon 2020 research and innovation programme under the Marie Skłodowska-Curie grant agreement No. 665778) and Poland's National Science Center "HARMONIA 3" grant 2012/06/M/ST10/00434. Numerical simulations were carried out at the Cyfronet AGH computer center, accessed through the PLGrid portal. We are grateful to Wojciech W.

Grabowski for fruitful discussions.

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
