# Peer review of "Stochastic coalescence in Lagrangian cloud microphysics"

_Atmospheric Chemistry and Physics, 2017_

## Referee Comment (RC1) · Anonymous Referee #1 · 15 May 2017

General comments:

The present paper describes the use of the super-droplet method (SDM) to study the collisional growth of cloud droplets. The technique is compared with DNS and the master equation. The results are new and original. The manuscript represents a good contribution to the development of new cloud microphysics models and is of potential interest for Atmospheric Chemistry and Physics community. Nevertheless, I have several comments/questions/suggestions that I hope will make this paper even more useful for the community.

Specific comments:

-The methodological section (2) should be expanded. From the current text, it is not possible to detect the equation of motions for the single droplets. Are the droplet tracers or inertial? Or are they just subjected to the gravitational force? A complete set of kinematic, dynamics and radius equation evolution should be given for a more general configuration, and then the system can be simplified depending on the hypothesis introduced by the authors.

-The English level of the manuscript needs to be improved. The most frequent error is the lack of articles in front of many substantive in all the manuscript (in collisional growth→ in the collisional growth, growth rate of lucky droplets → The growth rate of lucky droplets, just to give few examples). The origin derives from the lack of articles in Slavic languages, so the manuscript English level should be more carefully addressed in the next revision.

-The authors compare the DNS case by Onishi et al. (2015). It is not clear in the paper how the comparison has been done. Again, it is not clear if the super-droplets are influenced or not by (and if they move driven by) turbulent fluctuations or if the comparison is done just considering gravitational settling. The latter case would imply that the turbulent fluctuations have a weak effect on particle-particle collisions that it does not seem the case in reality.

-Many important references are missing regarding the methodology: Unterstrasser et al. 2016 doi:10.5194/gmd-2016-271, Li et al. 2017 doi:10.1002/2017MS000930. The latter, in particular, has many analogies and supports the results of the current manuscript.

-It would be interesting to include the effects of condensational growth as stated in the last paragraph of the conclusion. A new Lagrangian stochastic model has been proposed by Sardina et al. 2015 doi:10.1103/PhysRevLett.115.184501. The model could be easily implemented in the super-droplet framework.

-The LES sentence in the introduction can be obscure for non-specialist researchers in the field. The implication of the current approach for LES can be fundamental. The paragraph should be expanded to explain better the concept of LES and why collisions

**[ACPD](ACPD)**

should be accurately modelled in the absence of small turbulence scales.

-Section 6-Lucky droplets: The values of Kostinski and Shaw (2005) are "estimation". The sentence: "their theoretical analysis overestimates the luckiness in droplet growth" is too strong, the order of magnitude of their analysis is the same of the one detected with the super-droplet method.

Technical corrections:

-It is hard to distinguish the different lines in most of the plots if printed in black and white.

-For the reader point of view, it is easier if the comparison with the results of Alfonso and Raga (2016) are embedded directly in figure 1 and figure 2 (as the authors already did for figure 5).

-Figure 4: Is it possible to include in the plot the DNS results for a better comparison?

———————————————

---

## Referee Comment (RC2) · Anonymous Referee #2 · 24 May 2017

**Review of "*Stochastic coalescence in Lagrangian cloud microphysics?*" by Dziekan and Pawlowska**

Super droplet (SD) methods (SDM) are a novel and promising approach in cloud physics. Generally studies dealing with SD methods are welcome to better understand and gain experience with this type of model class. The present manuscript specifically addresses the coalescence process and compares a specific SD algorithm with other approaches like DNS, bin models or analytical derivations.
My impression is that the work is in general suitable for publication in ACP, as many interesting tests have been performed with a specific SDM.
Nevertheless the manuscript needs major revisions.
The presentation is too short to fully understand what has been done and to be able to judge it. Moreover, the statements are too general. Also the motivation behind choosing the presented tests must be made clearer. In the present style the presented tests resemble a bit a random collection of tests related to a specific SD coalescence algorithm.

**General points**

1. The description of your method and results is too short and often not clear enough. For a reviewer it is difficult to understand what you have done in detail and this makes it hard to thoroughly rate your work. The description in section 2 must be considerably expanded and be more precise. Also the results section should be improved. For example the motivation behind showing the comparison in section 3 is not really clear to me.

   As an example I go over page 3 and try to highlight what I miss there and where I suspect that other readers would get stuck:

   - It is not always clear if you talk about real droplets or super-droplets. It should be always clear if your statement relates to the real world or the super droplet world.
   - You should define the coalescence probability of a SD pair.
   - Can collections occur between droplets in one SD?
   - l.9: Do you use two types of simulations or is it only the initialisation that differs between the two types? Then I would not call it "types of simulations".
   - In general I can imagine how the "constant SD" initialisation works, however I am confused by your description.
     - l.21: do you mean large or small $N_{SD}$? Or large $r$?
     - $r_{min}$ and $r_{max}$ are defined by the relation in line 18. Using the $\Delta \ln(r)$ definition of line 19, you end up with implicit equations in the two variables. I am pretty sure this is not what you wanted to say.
     - You initialise additional SDs to better represent the tails of the SDs. Adding those SDs to your ensemble, isn't it necessary to reduce the multiplicity of the SD drawn from the largest bin? Is the refinement done at both sides? In the end, the actual number of SDs should be higher than the nominal value NSD? How much higher? Is the one-to-one initialisation the limiting case of the constant SD initialisation? Are the multiplicities in the constant SD approach integer values? How is the rounding done?
     - l.20: To be more explicit, you should write that one SD is created per bin and its radius is randomly selected.
     - l.23: Do you want to say that the probability *is allowed* to exceed 1. Why does this occur only here, and not in the one-to-one simulation? Is the time step longer? Why don't you reduce it then in this case?
     - What are multiple collections?
     - Do you use a constant time step throughout a simulation?

2. Your conclusions are too general. Your paper reads like a general comparison between the one and only SDM approach and all kinds of other algorithms.
Coalescence (or similarly coagulation of aerosols and dust or aggregation of ice crystals) can be treated in many ways in particle-based approaches (see algorithms by Andrejczuk et al, Shima et al, Riechelmann et al, Sölch & Kärcher, Zsom & Dullemond, Kolodko & Sabelfeld, deVille et al). There is not the one natural way to do this. Similarly, various algorithms exist for bin model approaches.
Please make clearer which statements hold in general and which are probably only valid for your specific SDM.
A recent study by Unterstrasser et al compared three different SD algorithms for the collection process (your all-or-nothing SDM is among the tested algorithms). Each algorithm has its strengths and weaknesses. One major finding was that the performance depends strongly on how the SDs are initialised from a given size distribution. This has several implications: 1. an accurate description of your initialisation is needed (see point 1) and 2. Are your simulations sensitive to initialisation details besides the number of SDs?

3. In my opinion you present five types of tests: 1. Comparisons with algorithms for the master equation (ME), 2. Comparisons with DNS algorithms 3. Comparisons with algorithms for the Smoluchowski equation (KCE) 4. Comparison with analytical results and 5. Stand-alone sensitivity tests of your SD algorithm.

With each type of test different aspects of your SD method can be tested. Each reference model you compare with (Bott, Alfonso, Onishi, Kostinski & Shaw) differs in which physical processes are explicitly treated. You often miss to clearly specify the purpose of and the motivation behind the individual evaluation steps you take. This must be made clearer in the revised version.

4. Your study is written in a style that implies that your SD algorithm does not solve the KCE. By comparing your SD results with KCE results you seemingly disclose shortcomings of the KCE description.
In fact, SDMs also rely on KCE and your study is not really suited to question the physics of the various formulations. In my understanding, you compare different numerical strategies of solving the KCE.
The probabilistic nature of your SDM is inherent to your specific algorithm. The SDMs of Riechelmann or Andrejczuk are not probabilistic. On the other hand, I agree that the probabilistic component of your SDM looks like it attempts to mimic the processes in nature. Nevertheless, the two should not be mistaken. In the superdropelt world of your SDM, the probabilistic approach comes from the fact, that each superdropelt represents a large number of real droplets and you may or may not perform a superdroplet collection. This not per se related to the stochastic nature of the real world processes that accounted for in the master equation.

**Major points:**

5. In my understanding there two mathematical descriptions of the coalescence process: The Smoluchowski equation and the master equation where only the latter accounts for correlations. You never state it explicitly but I suppose that your SD method is based on solving KCE.
Many spectral bin methods exist for the KCE and in the recent past SD methods for KCE were introduced. Hence, demonstrating agreement between your KCE solver and the ME solver by Alfonso is only reasonable for a physical problem where correlations are negligible.
If you want to show the numerical consistency of your approach, then a comparison with

other well-established KCE solvers would be more straightforward. Could you give reasons why you compare results of your KCE solver and an ME solver? Or do you want to claim that SDMs implicitly account for correlations?

6. I am not exactly sure what you intended to show in section 3. Did you want to demonstrate the suitability of your linear sampling of droplet pairs? Then you should compare results of your algorithm, once with linear sampling and once considering all possible combination. For a simulation with 30 droplets (as done in the comparison with Alfonso) this should be feasible. How do the curves in Fig.3 and 4 change, if you evaluate all possible combinations instead of linear sampling (at least for N0=10 and 100)?

7. Is it all reasonable trying to sample a continuous exponential distribution with 10 SDs? Are the total mass and number of the SDs (averaged over all ensembles) equal to the prescribed values? What about the higher moments, do they match the values of the analytical distribution? This is an important aspect as much of your evaluated variability may come from the initialisation and not so much from "stochasticity" of the SDM.

8. Even though Fig.4 shows simulations with an initialisation analogous to Onishi, the presented test is a stand-alone test (following the above categorisation). Hence, further tests with other kernels or other initial size distributions can be made (as no comparison with Onishi is required) in order to corroborate your findings about how many SDs are necessary to reach convergence.

9. In Fig.4 and following figures, does the Smoluchoswki line depend on the parameter choices of the underlying Bott algorithm (time step or number of bins)? Bott is known to be diffusive; this may explain the faster generation of large droplets.
   Anyway I would not call it the Smoluchowski line, it is the Bott line. The line may look differently for other bin KCE algorithms. Moreover, your lines are also "Smoluchowski" lines (see general point 4).

**Minor points:**

10. p.3, l21: Is the sol-gel transition an important aspect in cloud physics? You note that mass conservation is not guaranteed for some kernels. Are those kernels relevant in cloud physics? In super-droplet approaches mass conservation should be guaranteed by construction.

11. I do not understand the statement "the number of collision pairs is reduced" in p.4 l.4.

12. The paragraph starting p.4 l.11 should be moved to section 2.

13. For Figs.1 & 2 it would be ideal to obtain the data from Alfonso and include it in the plots.

14. P.2 l.13: Tanaka & Nakazawa present solutions for kernels other than the constant kernel. See also Table 2 in Alfonso, 2015.

15. P.5 l.5. Isn't this statement trivial? Probably any algorithm for KCE is faster than solving the master equation. A comparison among various KCE solvers would be fairer.

16. Figs. 2&3 show relative standard deviation of the largest droplet mass. Wouldn't it be interesting to know how large the largest droplet mass is? How many collections have occurred to form the largest droplet?

17. P.7 l.10-12: I am confused. DNS simulations compute droplet trajectories and directly evaluate if there are collisions between droplets. Why do you need a coalescence kernel in this numerical approach?

18. Fig.8: Do you also use the 20um cut-off in the Bott simulation? It is not explicitly stated in the text.

**Technical points:**
- Figure caption should include the information "t=2500s".
- P.5 l.9: droplet, not dropeltS.

**References:**

M. Andrejczuk, W. W. Grabowski, J. Reisner, und A. Gadian. Cloud-aerosol interactions for boundary layer stratocumulus in the Lagrangian cloud model. J. Geophys. Res., 115:D22214, 2010. doi: 10.1029/2010JD014248

R. DeVille, N. Riemer, und M. West. Weighted flow algorithms (WFA) for stochastic particle coagulation. J. Comput. Phys., 230(23):8427–8451, 2011. doi: 10.1016/j.jcp.2011.07.027.

A. Kolodko und K. Sabelfeld. Stochastic particle methods for Smoluchowski coagulation equation: Variance reduction and error estimations. Monte Carlo Methods and Applications, 9 (4):315–339, 2003. doi: 10.1163/156939603322601950

T. Riechelmann, Y. Noh, und S. Raasch. A new method for large-eddy simulations of clouds with Lagrangian droplets including the effects of turbulent collision. New Journal of Physics, 14(6):065008, 2012

S. Shima, K. Kusano, A. Kawano, T. Sugiyama, und S. Kawahara. The super-droplet method for the numerical simulation of clouds and precipitation: a particle-based and probabilistic microphysics model coupled with a non-hydrostatic model. Q. J. R. Meteorol. Soc., 135(642): 1307–1320, 2009.

I. Sölch und B. Kärcher. A large-eddy model for cirrus clouds with explicit aerosol and ice microphysics and Lagrangian ice particle tracking. Q. J. R. Meteorol. Soc., 136:2074–2093, 2010.

H. Tanaka und K. Nakazawa. Stochastic coagulation equation and validity of the statistical coagulation equation. Journal of Geomagnetism and Geoelectricity, 45(5):361–381, 1993. doi: 10.5636/jgg.45.361. URL http://doi.org/10.5636/jgg.45.361.

S. Unterstrasser, F. Hoffmann, und M. Lerch. Collection/aggregation algorithms in Lagrangian cloud microphysical models: rigorous evaluation in box model simulations. Geosci. Model Dev., 10(4):1521–1548, 2017. doi: 10.5194/gmd-10-1521-2017.

A. Zsom und C. Dullemond. A representative particle approach to coagulation and fragmentation of dust aggregates and fluid droplets. Astronomy and Astrophysics, 489(2):931–941, 2008. doi: 10.1051/0004-6361:200809921.

---

## Author Response (AR1)

**Reply to Reviewers**

July 14, 2017

We would like to thank the reviewers for their questions and comments. Before we answer them, we need to point out that we found an inconsistency in the way the collision efficiency tables were interpolated in the SDM and in the Bott method. It only affected simulations that use efficiencies from Hall (1980) for large droplets and from Davis (1972) for small droplets. This kind of collision kernel was used in Sections 5 and 6. The SDM simulations presented in Figs. 7, 8 and 9 were repeated with the problem fixed. The main difference is that now we see convergence of the "one-to-one" results to the Smoluchowski equation in the slow-coalescence case. What did not change is the fact that using larger coalescence cells can decrease the rate of conversion of cloud to rain drops due to additional collisions between rain drops. In consequence, using the Smoluchowski equation can underestimate the amount of rain produced. The problem affected only large drops, with radius greater than ca. 90 microns. Therefore the lucky droplet analysis from Sec. 6, in which droplets grow only up to 40 microns, remains valid.

**Answer to the Anonymous Referee #1.**

**- The methodological section (2) should be expanded. From the current text, it is not possible to detect the equation of motions for the single droplets. Are the droplet tracers or inertial? Or are they just subjected to the gravitational force? A complete set of kinematic, dynamics and radius equation evolution should be given for a more general configuration, and then the system can be simplified depending on the hypothesis introduced by the authors.**

We use box model simulations, which are convenient for studying coalescence. Droplet motion is not modelled, therefore we do not give their equations of motion. However, we use gravitational coalescence kernels, so droplets collide as if they settled due to gravitation. To clarify it, Section (2) was rewritten and says:

" Consider coalescence of water droplets in a well-mixed volume $V$. Other processes, like water condensation and evaporation, are not included. Thanks to the assumption that the volume is well-mixed, all droplets within the same well-mixed volume can collide with each other, independently of their positions (Gillespie, 1972). Therefore droplet motion does not have to be explicitly modeled and droplet coalescence can be calculated in a stochastic manner, as it is done in the master equation. Consider two randomly selected droplets $i$ and $j$. Probability that they collide during timestep $\Delta t$ is $P(r_i, r_j) = K(r_i, r_j)\Delta t/V$, where $r_i$ and $r_j$ are their radii, $K$ is the coalescence kernel and $V$ is volume of the box. We use gravitational coalescence kernels, so the effect of turbulence on coalescence is not studied. "

**- The English level of the manuscript needs to be improved. The most frequent error is the lack of articles in front of many substantive in all the manuscript (in collisional growth  in the collisional growth, growth rate of lucky droplets  The growth rate of lucky droplets, just to give few examples). The origin derives from the lack of articles in Slavic languages, so the manuscript English level should be more carefully addressed in the next revision.**

We made an effort to improve the text. If the manuscript is accepted, remaining errors will be fixed during the copy-editing that is included in the processing charges.

**- The authors compare the DNS case by Onishi et al. (2015). It is not clear in the paper how the comparison has been done. Again, it is not clear if the super-droplets are influenced or not by (and if they move driven by) turbulent fluctuations or if the comparison is done just considering gravitational settling. The latter case would imply that the turbulent fluctuations have a weak effect on particle-particle collisions that it does not seem the case in reality.**

We studied coalescence only due to gravitational settling. It is now written explicitly in Section 2:
"We use gravitational coalescence kernels, so the effect of turbulence on coalescence is not studied."
Onishi et al. (2015) performed DNS both for stagnant and turbulent air. They found that the mean autoconversion time is significantly decreased by turbulence, so turbulent fluctuations do have strong influence on collisions. However, we compare with them not the mean autoconversion time, but the relative standard deviation of autoconversion time and how it scales with the system size. Onishi et al. (2015) show that turbulence can change the relative standard deviation of autoconversion time by about 25%. While this is a significant change, it is barely visible on the logarithmic scale in Fig. 5.

**-Many important references are missing regarding the methodology: Unterstrasser et al.  2016 doi:10.5194/gmd-2016-271, Li et al.**

**2017 doi:10.1002/2017MS000930. The latter, in particular, has many analogies and supports the results of the current manuscript.**

Both papers are now cited in Section 1:

" A thorough comparison of coalescence algorithms from Lagrangian methods was done by Unterstrasser et al. (2016). It lead to the conclusion that the method of Shima et al. (2009) "yields the best results and is the only algorithm that can cope with all tested kernels". It was also found to be optimal in DNS tests (Li et al., 2017)."

Moreover, Li et al. (2017) is now cited in Section 8:

" Li et al. (2017) have shown that condensation can regulate differences between Eulerian and Lagrangian coalescence schemes. Discrepancies between these schemes that they observed in simulations with condensation and coalescence were smaller than in pure coalescence simulations. "

**-It would be interesting to include the effects of condensational growth as stated in the last paragraph of the conclusion. A new Lagrangian stochastic model has been proposed by Sardina et al. 2015 doi:10.1103/PhysRevLett.115.184501. The model could be easily implemented in the super-droplet framework.**

We have the option to include condensation in our model. We did not do it, because we believe that it is important to first understand the simpler problem of pure coalescence before dealing with more complex problems.

**-The LES sentence in the introduction can be obscure for non-specialist researchers in the field. The implication of the current approach for LES can be fundamental. The paragraph should be expanded to explain better the concept of LES and why collisions should be accurately modelled in the absence of small turbulence scales.**

We do not include turbulence in our coalescence scheme, so we think that there is no need to explain the concept of LES. We agree that the sentence could be obscure. Moreover, use of the super-droplet microphysics is not limited to LES. For these reasons we changed the sentence so that it does not mention LES anymore. Implications for cloud modeling, including LES, are discussed in Sec. 8.

**-Section 6-Lucky droplets: The values of Kostinski and Shaw (2005) are estimation. The sentence: their theoretical analysis overestimates the luckiness in droplet growth is too strong, the order of magnitude of their analysis is the same of the one detected with the super-droplet method.**

We changed that to:

"their theoretical analysis slightly overestimates the luckiness in droplet growth."

**Technical corrections: -It is hard to distinguish the different lines in most of the plots if printed in black and white.**

In addition to different colors, lines now also have different dashing.

**-For the reader point of view, it is easier if the comparison with the results of Alfonso and Raga (2016) are embedded directly in figure 1 and figure 2 (as the authors already did for figure 5).**

We have obtained the data from the authors of Alfonso and Raga (2016) and plotted it in the Figures 1 and 2.

**-Figure 4: Is it possible to include in the plot the DNS results for a better comparison?**

Onishi et al. (2015) give a DNS result for stagnant air only for one system size. We have added it to the Figure 4. The DNS result is significantly different from the SDM and the Smoluchowski equation results. Following Onishi et al. (2015), we conclude that this is due to the inaccuracy of the Hall coalescence kernel that was used in the latter two. Part of Section 4 that discusses Figure 4 now says:

" The SDM results are also compared with the results of DNS, in which air turbulence was not modelled, but hydrodynamic interactions between droplets were accounted for. We choose this kind of DNS, because it should be well described by the Hall kernel that is used in the SDM and in the Smoluchowski equation. It turns out that the Hall kernel gives too short autoconversion times. The same issue was observed by Onishi et al. (2015) (cf. Fig. 1(b) therein). "

Answer to the Anonymous Referee #2.

**- The presentation is too short to fully understand what has been done and to be able to judge it. Moreover, the statements are too general. Also the motivation behind choosing the presented tests must be made clearer. In the present style the presented tests resemble a bit a random collection of tests related to a specific SD coalescence algorithm.**

We have made an effort to make the statements and the presentation more specific. Regarding the motivation for different tests, we have rewritten the last paragraph to the introduction to make it more clear:

" The Shima algorithm is not based on the Smoluchowski equation, but, similarly to the master equation, on the assumption that the volume is well-mixed. The Shima algorithm introduces some simplifications that may increase the scale of fluctuations in the number of collisions, as described in Sec. 2. These simplifications are not necessary in the limiting case of a single computational particle representing a single real particle, what we call "one-to-one" simulations. Then, the Shima algorithm should be equivalent to the SSA, i.e. it should produce a realization in agreement with the master equation. To show that this is true, we compare the Shima algorithm with the master equation and the SSA in Sec. 3. We also compare it with the more fundamental DNS approach in Sec. 4. Once the "one-to-one" approach is shown to be at the same level of precision as the master equation, we use it to study some physical processes that are related to the stochastic nature of coalescence. The way the sol-gel transition time changes with system size is studied in Sec . 3 and in Sec. 6, we quantify how quickly the luckiest cloud droplets become rain drops. In addition, we use the "one-to-one" approach to validate more approximate methods. The Shima algorithm with multiplicities greater than one is studied in Sec. 4. We determine how many computational particle s are required to obtain the correct mean autoconversion time and correct fluctuations in the autoconversion time. Next, in Sec. 5, we determine how large the system has to be for the Smoluchowski equation to correctly represent the rate of rain formation. Throught the paper we observe that evolution of the system strongly depends on its size. The size of a well-mixed air parcel is estimated in Sec. 7 and some implications for cloud simulations are discussed in Sec. 8. "

In addition, we changed the titles of Secs. 3 and 4 to make it more clear what is their purpose.

**General points 1. The description of your method and results is too short and often not clear enough. For a reviewer it is difficult to understand what you have done in detail and this makes it hard to thoroughly rate your work. The description in section 2 must be considerably expanded and be more precise. Also the results section should be improved. For example the motivation behind showing the comparison in section 3 is not really clear to me. As an example I go over page 3 and try to highlight what I miss there and where I suspect that other readers would get stuck:**

The method was described in detail by Shima et al. (2009), so our intention was only to describe how our simulation method differs from theirs. Apparently this makes the method not clear, so we extend the description of the method as asked by the Reviewer in the following points. Regarding the motivation behind different comparisons, it is now given in the Introduction, as explained in the answer to the previous comment.

**It is not always clear if you talk about real droplets or super-droplets. It should be always clear if your statement relates to the real world or the super droplet world.**

It is now clarified in Sec. 1:

" We will refer to these computational particles as super-droplets (SDs). The words "droplets" and "drops" are reserved for real hydrometeors. "

**You should define the coalescence probability of a SD pair.**

It is now given in Sec. 2: " Probability of coalescence of two SDs $i$ and $j$ that belong to the same collision pair is $P_{SD}(r_i, r_j, \xi_i, \xi_j) = \max(\xi_i, \xi_j)P(r_i, r_j)(N_{SD}(N_{SD}-1)/2)/\lfloor N_{SD}/2 \rfloor$ (Shima et al., 2009).
"

**Can collections occur between droplets in one SD?**

We use gravitational coalescence kernels, so droplets in one SD all have the same terminal velocity and therefore cannot collide. We now say it explicitly in Sec. 2:

" Real droplets represented by the same SD cannot collide with each other, because they have the same sedimentation velocities. "

**l.9: Do you use two types of simulations or is it only the initialisation that differs between the two types? Then I would not call it types of simulations.**

We use two types of simulations. The main difference, besides the initialisation, is that in the "one-to-one" simulations the timestep is adaptive, as stated in Sec. 2:

"Timestep length is adapted at each step to ensure that none of the collision pairs has coalescence probability greater than one."

In the "constant SD" simulations the timestep is constant and multiple collisions between SDs in a single timestep are allowed, as said in Sec. 2:

" In this type of simulation, the time step length is constant $\Delta t = 1$ s. It is not adapted, as it is done in the "one-to-one" simulations, to make the simulation computationally more efficient. Using constant time step length can make the coalescence probability exceed unity. If it does, it represent multiple collisions between a pair of SDs (Shima et al., 2009). "

**In general I can imagine how the constant SD initialisation works, however I am confused by your description. o l.21: do you mean large NSD or small NSD? Or large r?**

As written, we mean large $N_{SD}$. For large $N_{SD}$, $\Delta l_r$ becomes small and therefore $r_{max}$ is small.

**o rmin and rmax are defined by the relation in line 18. Using the ln(r) definition of line 19, you end up with implicit equations in the two variables. I am pretty sure this is not what you wanted to say.**

Yes, we end up with implicit equations for these variables. Many pairs of values of $r_{min}$ and $r_{max}$ could satisfy them. We find our solution numerically, what is now explained in more detail in Sec. 2:

" The first step of the initialization is finding the largest and smallest initial super-droplet radius, $r_{max}$ and $r_{min}$. They are found iteratively, starting with $r_{min} = 10^{-9}$ m and $r_{max} = 10^{-3}$ m. We require that they satisfy the condition

$$n(\ln(r_e))\Delta l_r V \geq 1, \tag{1}$$

where $r_e$ is either $r_{max}$ or $r_{min}$, $n(\ln(r))$ is the initial droplet size distribution and $\Delta l_r = (\ln(r_{max}) - \ln(r_{min}))/N_{SD}$. In each iteration, if $r_{min}$ ($r_{max}$) does not satisfy (1), it is increased (decreased) by 1%. "

**o You initialise additional SDs to better represent the tails of the SDs. Adding those SDs to your ensemble, isnt it necessary to reduce the multiplicity of the SD drawn from the largest bin?**

No, in "constant SD" the right edge of the largest bin is $r_{max}$, so effectively the distribution is cut at $r_{max}$ and droplets with $r > r_{max}$, that would be present in the real system, are not accounted for. Adding additional SDs with $r > r_{max}$ fixes this problem and does not affect the number of droplets with $r \leq r_{max}$ (i.e. the multiplicity of SDs with $r \leq r_{max}$).

**Is the refinement done at both sides?**

No, only on the large radius side. We now write explicitly in Sec. 2:
" We do not add SDs from the small tail of the distribution, because very small droplets are of little importance for rain formation. "

**In the end, the actual number of SDs should be higher than the nominal value NSD? How much higher?**

Yes, it is a little higher, what is now written in Sec. 2:
" This makes the actual number of SDs higher than the prescribed value $N_{SD}$, typically by ca. 1%. "

**Is the one-to-one initialisation the limiting case of the constant SD initialisation?**

No, because using $N_{SD} = N_0$ (i.e. multiplicity = 1) in a "constant SD" simulation would result in relatively small $r_{max}$ and large $r_{min}$. Then this type of initialisation would not represent well the given distribution. Therefore a different approach to initialisation is used in the "one-to-one" simulations.

**Are the multiplicities in the constant SD approach integer values? How is the rounding done?**

Yes they are, as explained in Shima et al. (2009). The rounding is done to the nearest integer. The error introduced is small, because multiplicities are high.

o **l.20: To be more explicit, you should write that one SD is created per bin and its radius is randomly selected.**

It is written in Sec. 2:
" Once $r_{min}$ and $r_{max}$ are found, radius of one SD is randomly selected within each bin of size $\Delta l_r$. "

o **l.23: Do you want to say that the probability is allowed to exceed 1. Why does this occur only here, and not in the one-to-one simulation? Is the time step longer? Why dont you reduce it then in this case?**

Yes, in the method of Shima et al. (2009) the coalescence probability is allowed to exceed 1. It is a consequence of keepeing the time step length constant. We added to Sec. 2:
" In this type of simulation, the time step length is constant $\Delta t = 1$ s. It is not adapted, as it is done in the "one-to-one" simulations, to make the simulation computationally more efficient. Using constant time step length can make the coalescence probability exceed unity. If it does, it represent multiple collisions between a pair of SDs (Shima et al., 2009). "

o **What are multiple collections?**

It means that, during a single time step, a pair of SDs collides more than once. The details are given in Shima (2009).

o **Do you use a constant time step throughout a simulation?**

In the "one-to-one" simulations it is adaptive, as said in Sec. 2:
" Time step length is adapted at each time step to ensure that none of the collision pairs has coalescence probability greater than one. "
In the "constant SD" simulations it is constant, what is now explicitly stated in Sec. (2):
" In this type of simulations, the time step length is constant $\Delta t = 1$ s. "

**2. Your conclusions are too general. Your paper reads like a general comparison between the one and only SDM approach and all kinds of other algorithms. Coalescence (or similarly coagulation of aerosols and dust or aggregation of ice crystals) can be treated in many ways in particle-based approaches (see algorithms by Andrejczuk et al, Shima et al, Riechelmann et al, Solch & Karcher, Zsom & Dullemond, Kolodko & Sabelfeld, deVille et al). There is**

**not the one natural way to do this. Similarly, various algorithms exist for bin model approaches. Please make clearer which statements hold in general and which are probably only valid for your specific SDM.**

Regarding coalescence, the main difference between these particle-based approaches (with the exception of the DeVille algorithm, which is based on the Smoluchowski equation) is in what is the outcome of a collision of super-droplets with multiplicities $\xi > 1$. Majority of our simulations were the "one-to-one" simulations, in which $\xi = 1$. In that case, it is straightforward what the result of a collision should be, so differences between these algorithms disappear. Moreover, like the master equation, the "one-to-one" simulations are only based on the assumption that the cell is well-mixed. The numerical trick of reducing the number of collision pairs ("linear sampling") does not affect the fluctuations, as we show in Sec. 3. Therefore the "one-to-one" simulations are quite similar to the SSA. They are at the same level of accuracy as the master equation: less precise than the DNS, more precise than the Smoluchowski equation. It is now cleary stated in the Introduction:

" The Shima algorithm is not based on the Smoluchowski equation, but, similarly to the master equation, on the assumption that the volume is well-mixed. The Shima algorithm introduces some simplifications that may increase the scale of fluctuations in the number of collisions, as described in Sec. 2. These simplifications are not necessary in the limiting case of a single computational particle representing a single real particle, what we call "one-to-one" simulations. Then, the Shima algorithm should be equivalent to the SSA, i.e. it should produce a realization in agreement with the master equation. To show that this is true, we compare the Shima algorithm with the master equation and the SSA in Sec. 3. We also compare it with the more fundamental DNS approach in Sec. 4. "

In simulations with $\xi > 1$, we use the Shima method, as it was found to be optimal by Unterstrasser et al. (2017) and by Li et al. (2017). These simulations are used only in Sec. 4 in order to determine their accuracy, as explained by an added paragraph in the introduction:

" In addition, we use the "one-to-one" approach to validate more approximate methods. The Shima algorithm with multiplicities greater than one is studied in Sec. 4. We determine how many computational particles are required to obtain the correct mean autoconversion time and correct fluctuations in the autoconversion time. "

**A recent study by Unterstrasser et al compared three different SD algorithms for the collection process (your all-or-nothing SDM is among the tested algorithms). Each algorithm has its strengths and weaknesses. One major finding was that the performance depends strongly on how the SDs are initialised from a given size distribution. This has several implications: 1. an accurate description of your initialisation is needed (see point 1) and**

Unterstrasser et al. (2017) found that the Shima method is optimal, what we now say in the introduction: " A thorough comparison of coalescence algorithms from Lagrangian methods was done by Unterstrasser et al. (2017). It lead to the conclusion that the method of Shima (2009) "yields the best results and is the only algorithm that can cope with all tested kernels". It was also found to be optimal in DNS tests Li et al. (2017). In the light of these results, we choose to use the coalescence algorithm of Shima (2009) in this work. "

We also observed that the way the initialisation is done is important. We believe that now our initialisation algorithm is described more clearly.

**2.Are your simulations sensitive to initialisation details besides the number of SDs?**

It is only sensitive to the number of SDs. Initial values of $r_{min}$ and $r_{max}$, if reasonable, do not affect it much.

**3. In my opinion you present five types of tests: 1. Comparisons with algorithms for the master equation (ME), 2. Comparisons with DNS algorithms 3. Comparisons with algorithms for the Smoluchowski equation (KCE) 4. Comparison with analytical results and 5. Stand-alone sensitivity tests of your SD algorithm. With each type of test different aspects of your SD method can be tested. Each reference model you compare with (Bott, Alfonso, Onishi, Kostinski & Shaw) differs in which physical processes are explicitly treated. You often miss to clearly specify the purpose of and the motivation behind the individual evaluation steps you take. This must be made clearer in the revised version.**

We believe that it is now made clear by the following paragraph in the introduction:

" The Shima algorithm is not based on the Smoluchowski equation, but, similarly to the master equation, on the assumption that the volume is well-mixed. The Shima algorithm introduces some more simplifications that may increase the scale of fluctuations in the number of collisions, as described in Sec. 2. These additional simplifications are not necessary in the limiting case of a single computational particle representing a single real particle, what we call "one-to-one" simulations. Then, the Shima algorithm should be equivalent to the SSA, i.e. it should produce a trajectory in agreement with the master equation. To show that this is true, we compare the Shima algorithm with the master equation and the SSA in Sec. 3. We also compare it with the more fundamental DNS approach in Sec. 4. Once the "one-to-one" approach is shown to be at the same level of precision as the master equation, we use it to study some physical processes that are related to the stochastic nature of coalescence. The way the sol-gel transition time changes with system size is studied in Sec. 3 and in Sec. 6, we quantify how quickly the luckiest cloud droplets become rain drops. In addition, we use the "one-to-one" approach to validate more

approximate methods. The Shima algorithm with multiplicities greater than one is studied in Sec. 4. We determine how many computational particles are required to obtain the correct mean autoconversion time and correct fluctuations in the autoconversion time. Next, in Sec. 5, we determine how large the system has to be for the Smoluchowski equation to correctly represent the rate of rain formation. "

**4. Your study is written in a style that implies that your SD algorithm does not solve the KCE. By comparing your SD results with KCE results you seemingly disclose shortcomings of the KCE description. In fact, SDMs also rely on KCE and your study is not really suited to question the physics of the various formulations. In my understanding, you compare different numerical strategies of solving the KCE.**

The SD algorithm is not a method of solving the KCE. Contrary to the KCE, it does include correlations between number of droplets of different sizes. The "one-to-one" simulations are similar to the SSA, i.e. they produce a single trajectory that follows the master equation (c.f. Sec. 2). As such, they are more precise than the KCE, so they are well-suited to disclose shortcomings of the KCE. To us it is not clear how the "constant SD" simulations relate to the KCE and the master equation. They have been shown to give mean result in agreement with the KCE in large systems (Shima et al., 2009, Unterstrasser et al., 2016). Regarding fluctuations, using the "all-or-nothing" algorithm should amplify fluctuations, because it introduces unrealistic correlations between number of droplets of different sizes. In Sec. 4 we quantify how much the fluctuations amplitude increases.

**The probabilistic nature of your SDM is inherent to your specific algorithm. The SDMs of Riechelmann or Andrejczuk are not probabilistic. On the other hand, I agree that the probabilistic component of your SDM looks like it attempts to mimic the processes in nature. Nevertheless, the two should not be mistaken. In the superdropelt world of your SDM, the probabilistic approach comes from the fact, that each superdropelt represents a large number of real droplets and you may or may not perform a superdroplet collection. This not per se related to the stochastic nature of the real world processes that accounted for in the master equation.**

The probabilistic nature of our SDM has the same source as the probabilistic nature of the master equation, i.e. the fact that a collision between a pair of SDs happens with some probability, according to the assumption that the volume is well-mixed. To our knowledge, this Monte Carlo approach to collisions is used in all SDMs, including the ones of Riechelmann and Andrejczuk. If multiplicities are equal to one ("one-to-one" simulations), the SDM is as much related to the real world process as the master equation. If multiplicities are greater than one,

various SDM algorithms start to differ. In the Shima algorithm that we use, the scale of fluctuations is increased, because the number of collision trials is lower than it would be in reality. We try to quantify how much it is increased. In the Riechelmann and Andrejczuk SDMs, the fluctuations are lower than in the one of Shima. Unfortunately, they do not give mean results as good as the Shima algorithm (Unterstrasser et al., 2016).

**Major points: 5. In my understanding there two mathematical descriptions of the coalescence process: The Smoluchowski equation and the master equation where only the latter accounts for correlations. You never state it explicitly but I suppose that your SD method is based on solving KCE. Many spectral bin methods exist for the KCE and in the recent past SD methods for KCE were introduced. Hence, demonstrating agreement between your KCE solver and the ME solver by Alfonso is only reasonable for a physical problem where correlations are negligible. If you want to show the numerical consistency of your approach, then a comparison with other well-established KCE solvers would be more straightforward. Could you give reasons why you compare results of your KCE solver and an ME solver? Or do you want to claim that SDMs implicitly account for correlations?**

Our SD method is not based on solving KCE. The "one-to-one" simulations are at the level of precision of the master equation, what is now explicitly written in the introduction:

" The Shima algorithm is not based on the Smoluchowski equation, but, similarly to the master equation, on the assumption that the volume is well-mixed. The Shima algorithm introduces some more simplifications that may increase the scale of fluctuations in the number of collisions, as described in Sec. 2. These additional simplifications are not necessary in the limiting case of a single computational particle representing a single real particle, what we call "one-to-one" simulations. Then, the Shima algorithm should be equivalent to the SSA, i.e. it should produce a trajectory in agreement with the master equation. To show that this is true, we compare the Shima algorithm with the master equation and the SSA in Sec. 3. "

We compare "one-to-one" simulations with the master equation to validate the claim that they are at the same level of precision, i.e. that "one-to-one" method accounts for correlations. In the problem of Alfonso, correlations are very important and, as shown in Alfonso and Raga (2017), the KCE does not solve it well.

**6. I am not exactly sure what you intended to show in section 3. Did you want to demonstrate the suitability of your linear sampling of droplet pairs? Then you should compare results of your algorithm, once with linear sampling and once considering all possible combination. For a simulation with 30 droplets (as done in the comparison**

**with Alfonso) this should be feasible. How do the curves in Fig.3 and 4 change, if you evaluate all possible combinations instead of linear sampling (at least for N0=10 and 100)?**

We intended to show that "one-to-one" simulations agree with the master equation. Linear sampling is an optimization technique that we expected might be responsible for some differences between the master equation and "one-to-one" simulations. To make the comparison more detailed, now in Section 3 we compare "one-to-one" simulations, with and without linear sampling, with the master equation. We find that linear sampling does not affect mean number of collisions, nor the fluctuations in the number of collisions. Figs. 3 and 4 (up to N0=100) do not change if linear sampling is not used.

The second paragraph of Section 3 has been rewritten to explain these new results of simulations without linear sampling.

**7. Is it all reasonable trying to sample a continuous exponential distribution with 10 SDs? Are the total mass and number of the SDs (averaged over all ensembles) equal to the prescribed values? What about the higher moments, do they match the values of the analytical distribution? This is an important aspect as much of your evaluated variability may come from the initialisation and not so much from stochasticity of the SDM.**

Averaged over the ensamble, up to the 4-th moment of the distribution is in agreement with the prescribed one. It is true that the initial distributions can be very different between realisations and it may be the cause of large variability. For this reason we removed the $N_0 = 10$ case from Fig. 3.

**8. Even though Fig.4 shows simulations with an initialisation analogous to Onishi, the presented test is a stand-alone test (following the above categorisation). Hence, further tests with other kernels or other initial size distributions can be made (as no comparison with Onishi is required) in order to corroborate your findings about how many SDs are necessary to reach convergence.**

Now in Fig. 4 we compare our results with Onishi's results. Our expectation is that for other kernels relevant for cloud physics the results would not be much different. Nevertheless, we agree that tests with other kerneles, and with other initial conditions, would be useful. Such tests could easily fill a whole new paper. Our result can be considered as a guideline for users, who should do convergence tests for the specific kernels they use.

**9. In Fig.4 and following figures, does the Smoluchoswki line depend on the parameter choices of the underlying Bott algorithm (time step or number of bins)? Bott is known to be diffusive; this may explain the faster generation of large droplets. Anyway I would**

**not call it the Smoluchowski line, it is the Bott line. The line may look differently for other bin KCE algorithms. Moreover, your lines are also Smoluchowski lines (see general point 4).**

We have done convergence tests of the Bott algorithm. It is now explained in Sec. 4:

" In the Bott algorithm, we used $\Delta t = 1$ s and mass bin spacing $m_{i+1} = 2^{1/10} m_i$. The same parameters were used in each simulation presented in this manuscript. Convergence tests were done for each case. "

We agree that the Bott algorithm produces rain too soon most probably due to numerical diffusion. We now write in Sec. 4:

" The "one-to-one" results converge with increasing system volume (i.e. increasing $N_0$) to a value higher than the Smoluchowski result. The difference is probably caused by the numerical diffusion of the Bott algorithm. "

Labels on figures are one of: DNS, master equation, SSA, SDM ("one-to-one" or "constant SD"), Smoluchowski equation. In our view these are different appproches to solving droplet coalescence, not different numerical methods for solving some equation. The numerical methods used, i.e. the Bott algorithm for the Smoluchowski equation and the Alfonso algorithm for the master equation, are explained in text.

**Minor points: 10. p.3, l21: Is the sol-gel transition an important aspect in cloud physics? You note that mass conservation is not guaranteed for some kernels. Are those kernels relevant in cloud physics? In super-droplet approaches mass conservation should be guaranteed by construction.**

Mass is not conserved for the multiplicative kernel, which is not relevant in cloud physics. Nevertheless, the paper Alfonso and Raga (ACP, 2017) is a detailed study of the sol-gel transition in a small cloud volume. We decided that it will be interesting to extend their results to more realistic cases.

**11. I do not understand the statement the number of collision pairs is reduced in p.4 l.4.**

It was supposed to mean that linear sampling is used, i.e. less collision pairs are considered than in an exact description. Now in Sec. 2 we define the meaning of linear sampling, which is later used in the paper:

" The second simplification, that we will refer to as linear sampling, is that instead of considering all $N_{SD}(N_{SD}1)/2$ collision pairs, only $[N_{SD}/2]$ non-overlapping pairs are randomly selected. "

**12. The paragraph starting p.4 l.11 should be moved to section 2.**

In the paragraph it was shown that in the "one-to-one" method with linear sampling, the probability of collision between any two real droplets is the same

as in simulations without linear sampling. This does not ensure that fluctuations in the number of collisions are also correctly represented. Therefore we removed the paragraph. In its place we added to Figs. 1 and 2 the results of simulations without linear sampling, i.e. with all collision pairs considered. Their agreement with the linear sampling simulations implies that linear sampling does not affect the scale of fluctuations. Proving that was the point of the removed paragraph.

**13. For Figs.1 & 2 it would be ideal to obtain the data from Alfonso and include it in the plots.**

We did that.

**14. P.2 l.13: Tanaka and Nakazawa present solutions for kernels other than the constant kernel. See also Table 2 in Alfonso, 2015.**

We now cite Tanaka and Nakazawa in the introduction:
" The master equation was analytically solved only for monodisperse initial conditions with simple coalescence kernels (Bayewitz et al., 1974; Tanaka and Nakazawa, 1993). "
Table 2 in Alfonso (2013) gives solutions of the Smoluchowski equation, not the master equation.

**15. P.5 l.5. Isnt this statement trivial? Probably any algorithm for KCE is faster than solving the master equation. A comparison among various KCE solvers would be fairer.**

As explained previously, SDM is similar to the SSA and not to KCE solvers. For this reason we compare with the SSA and a solver of the master equation.

**16. Figs. 2&3 show relative standard deviation of the largest droplet mass. Wouldnt it be interesting to know how large the largest droplet mass is? How many collections have occurred to form the largest droplet?**

Relative standard deviation of the largest droplet mass is interesting as a measure of the sol-gel transition. We do not see a reason to show the mass of it or the number of collisions that lead to it.

**17. P.7 l.10-12: I am confused. DNS simulations compute droplet trajectories and directly evaluate if there are collisions between droplets. Why do you need a coalescence kernel in this numerical approach?**

It was our error. The DNS was done not for different coalescence kernels, but for different turbulence strength. We changed that sentence to:
" Small discrepancies are probably caused by the fact that the DNS included turbulence of various strength for different $N_0$. "

**18. Fig.8: Do you also use the 20um cut-off in the Bott simulation? It is not explicitly stated in the text.**

Yes, we do use it in the Bott simulation. It is now said explicitly:

" In addition, we cut the distribution to 0 at $r = 20$ m. This cutoff is used in SDM modelling as well as when solving the Smoluchowski equation. "

**Technical points: Figure caption should include the information t=2500s.**

We add it to the caption of Fig. 1.

**P.5 l.9: droplet, not dropeltS.**

Fixed.

[revised manuscript text omitted]
, but the number of collision pairs is reduced. Figure 1 shows the mass distribution between droplet sizes after $t = 2500$ s averaged over an ensemble of $\Omega = 10^5$ realizations. It compares well with results presented in Fig. 8 in Alfonso and Raga (2016). This implies that the "~~" simulations are compared with the master equation approach. Both approaches are generally in agreement, with some differences at the large end of the distribution.

[Figure]

**Figure 1.** Mass of droplets per size bin at $t = 2500$ s. Bins are 1 μm wide. Points depict an averaged result of $\Omega = 10^4$ "one-to-one" simulations with and without linear sampling of collision pairs. Error bars show a 95% confidence interval. Line depicts a numerical solution of the master equation (see Fig. 8 in Alfonso and Raga (2017), data courtesy of L. Alfonso).

These differences may be caused by the way how the coalescence efficiency tables are interpolated. Another possible source of discrepancies is the numerical diffusion present in the finite-differences method of Alfonso (2015). To test if the "one-to-one"  " method also gives correct fluctuations in the number of collisions, relative standard deviation of mass

5 of the largest droplet $\sigma(m_{max})/\langle m_{max}\rangle$ is plotted in Fig. 2.  "One-to-one" simulations, with and without linear sampling, are compared with SSA simulations. As in Fig. 1, we do not observe any negative effect of using the linear sampling technique and the "one-to-one" simulations compare relatively well with the SSA. Possible sources if discrepancies are the same as in Fig. 1. Judging from

10 Figs. 1 and 2, we conclude that the "one-to-one" approach is in agreement with the master equation approach. It accounts for the correlations in the number of droplets per size-bin and as such is more fundamental than the Smoluchowski equation approach.

[Figure]

**Figure 2.** Relative standard deviation of mass of the largest droplet in the system. Details of the SDM simulations are given in the caption of Fig. 1. Size of the ensemble of SSA simulations is $\Omega = 10^3$. The SSA results are taken from Fig.7 in Alfonso and Raga (2017) (data courtesy of L. Alfonso).

~~$\eta = \frac{N_{SD}(N_{SD}-1)}{2}/\left[\frac{N_{SD}}{2}\right]$ is the scaling-up of probability (Shima et al., 2009) and $P_{col}dt$ is the probability of coalescence if all pairs were considered. To calculate $P_{pair}$, we first consider even values of $N_{SD}$. Consider a random permutation of droplet indices. Probability that the first droplet from the pair is at an odd position in the permutation and the second is at the next position to the right is $\frac{1}{2}\frac{1}{N_{SD}-1}$. Probability that the first is at an even position and the second is to the left of it is the same. Summing these two we get $P_{pair}^{even}(N_{SD}) = 1/(N_{SD}-1)$. If $N_{SD}$ is odd, the probability is $P_{pair}^{odd} = P_{pair}^{even}(N_{SD}-1)\frac{N_{SD}-2}{N_{SD}}$. We can write an expression for both odd and even cases $P_{pair} = 1/(N_{SD}-1+2*(N_{SD}/2-[N_{SD}/2]))$. It is readily obtained that $P_o = P_{col}$, i.e. that the probability of collision between any pair of real droplets is conserved in the "~~

The "one-to-one"

" SDM with linear sampling is computationally more efficient than solving the master equation directly, or using the SSA. It also puts no constraints on the initial distribution of droplets. Therefore we can use SDM to predict gelation times for larger systems and more realistic initial conditions. We use an initial droplet distribution that is exponential in mass $n(m) = \frac{n_0}{\overline{m}}exp(-m/\overline{m})$, where $n(m)dm$ is the number of droplets in mass range $(m, m+dm)$ in unit volume, $n_0 = 142$ cm$^{-3}$ and $\overline{m}$ is the mass of a  droplet with radius $\overline{r} = 15$ μm. This is the same distribution as in  Onishi et al. (2015). The total initial number of droplets in the system is $N_0 = n_0 V$. Results of the  "oneto-one"" simulations for $N_0$ up to  $10^5$ are shown in Fig. 3. For $N_0 \geq 10^2$, the relative standard deviation of mass of the largest droplet, which quantifies amplitude of fluctuations, decreases with increasing system size. This can be understood if we look at a larger cell as an ensemble of ten smaller cells. Comparing between independent realizations, variability in the size of the single, largest droplet will be smaller if this droplet is selected from ten cells in each realization than if it was selected from only a single cell per realization. Interestingly, for $N_0 = 10^5$ an inflection point appears around $t = 500$ s. It is not seen in smaller cells. This indicates that some new source of variability is introduced. We believe that it is associated with collisions between large rain drops. We will come back to this in Sec. 5.

The sol-gel transition time coincides with the time at which $\sigma(m_{max})/\langle m_{max} \rangle$ reaches maximum  Intuitively, we would expect the time for most of the mass to accumulate in a single agglomerate to increase with increasing system size. This turns out to be true for systems with $N_0 > 10^3$. For system sizes $10^2 < N_0 < 10^3$ gelation time is approximately the same, around 300 s. ~~Behavior of an extremely small system with only 10 droplets is much different. Maximum relative fluctuations are smaller and gelation time is longer than in a ten times larger system. Also, the maximum of $\sigma(m_{max})/\langle m_{max} \rangle$ is not very distinct. This is a manifestation of strong correlations in number of droplets of a given size. For example, if particles collide to form only two droplets of similar size, these two droplets may not collide for a very long time. Hence we observe large fluctuations even at $t = 2500$ s.~~

[revised manuscript text omitted]
. ~~In box model simulations, the Smoluchowski equation produces too much rain if initial distribution is well below size gap and droplets slowly grow through coalescence. It is difficult to tell if using the Smoluchowski equation in cloud models overestimates the amount of rain. Possibly, condensational growth helps droplets cross the gap, leading to initial condition that is closer to the one in the first case ($\bar{r} = 15$ μm). For such initial condition, Smoluchowski equation gives correct results.~~

**6 Lucky droplets**

There is a well-established idea that some droplets undergo series of unlikely collisions and grow much faster than an average droplet (Telford, 1955; Scott, 1967; Marcus, 1968; Robertson, 1974; Mason, 2010). These few lucky droplets are

[Figure]

**Figure 9.** Mean concentration of rain drops from the same simulations as in Fig. 8.

**Table 1.** Average, standard deviation and sample size of time (in seconds) for the lucky realizations to produce single rain drop.

| $10^{-4}$ | | $\gamma = 10^{-3}$ | | | $\gamma = 10^{-2}$ | | | $\gamma = 10^{-1}$ | | | |
|---|---|---|---|---|---|---|---|---|---|---|---|
| $\langle t^\gamma_{40}\rangle\text{-}\sigma(t_{40})_\gamma$ | $\gamma\Omega$ | $\langle t^\gamma_{40}\rangle\text{-}\langle t_{40}\rangle_\gamma$ | $\sigma(t^\gamma_{40})\text{-}\sigma(t_{40})_\gamma$ | $\gamma\Omega$ | $\langle t^\gamma_{40}\rangle\text{-}\langle t_{40}\rangle_\gamma$ | $\sigma(t^\gamma_{40})\text{-}\sigma(t_{40})_\gamma$ | $\gamma\Omega$ | $\langle t^\gamma_{40}\rangle\text{-}\langle t_{40}\rangle_\gamma$ | $\sigma(t^\gamma_{40})\text{-}\sigma(t_{40})_\gamma$ | $\gamma\Omega$ | $\langle t^\gamma_{40}\rangle\text{-}\langle t_{40}\rangle_\gamma$ |
| 212 | 10 | 2930 | 356 | 10 | 4053 | 517 | $10^2$ | 6365 | 1158 | $10^3$ | 14777 |
| 120 | $10^2$ | 1762 | 170 | $10^3$ | 2400 | 267 | $10^4$ | 3440 | 505 | $10^5$ | 6500 |
| 173 | 3 | 1336 | 103 | 10 | 1717 | 176 | $10^2$ | 2354 | 276 | $10^3$ | 3912 |
| 33 | 2 | 1090 | 60 | 20 | 1334 | 85 | 200 | 1721 | 169 | 2000 | 2552 |
| | | | | | 1038 | 165 | 2 | 1301 | 176 | 20 | 1831 |

argued to be responsible for droplet spectra broadening and rain  forming quicker than predicted by the Smoluchowski equation. Luck is supposed to be especially important during crossing of the size gap, when collisions happen rarely (Robertson, 1974; Kostinski and Shaw, 2005). A single droplet that would cross the size gap through lucky collisions could then initiate a cascade of collisions.

5  Theoretical estimation of the  "luck factor" was presented in Kostinski and Shaw (2005). We use  "one-to-one" simulations to test predictions from that paper.

We are interested in time $t_{40}$ it takes for the largest droplet in the system to grow to $r = 40$ μm. We perform simulations for the same initial distribution as in the second case in Sec. 5. The mean radius is $\bar{r} = 9.3$ μm, well below the size gap. From an ensemble of $\Omega$ realizations, we select sub-ensembles of luckiest realizations, i.e. those with the smallest $t_{40}$. We consider sub-ensembles of size $\gamma\Omega$ with $log_{10}(\gamma) = -4, -3, -2, -1, 0$. In each sub-ensemble, we calculate  the mean $\langle t_{40}\rangle_\gamma$ and the standard deviation $\sigma(t_{40})_\gamma$, where the subscript $\gamma$ denotes the size of the sub-ensemble from which the statistic was calculated. The results for different cell sizes are shown in Tab. 1. There is a large variability in  $\langle t_{40}\rangle_\gamma$ with cell size. This is caused by the fact that $t_{40}$ depends only on a single largest droplet. Larger cells contain more droplets, so probability of producing single large droplet increases with cell size. We notice that  $\langle t_{40}\rangle_\gamma$ is approximately the same along the diagonals of Tab. 1. For example, cell containing $10^6$ droplets on average will produce first rain droplet in 30 minutes. If we divided it into 10 cells with $10^5$ droplets each, the luckiest one would also produce a droplet in 30 minutes on average. This shows that using large coalescence cells does not affect formation of first rain drops. The differences discussed in previous Sections emerge later, when there are already some rain drops that can collide with each other. Moving to very small cells, we no longer observe same  $\langle t_{40}\rangle_\gamma$ along the diagonals. Ten cells with $N_0 = 10^2$ produce rain drops slower than a single cell with $N_0 = 10^3$. This is due to depletion of water droplets in small cells. The largest droplet a cell with $N_0 = 10^2$ can produce has $r \approx 43$ μm, close to the 40 μm rain threshold.

 Kostinski and Shaw (2005) estimate that the luckiest $10^{-3}$ fraction of droplets should cross the size gap around six times faster than average, while the luckiest $10^{-5}$ around nine times faster. We compare these values with our simulations for $N_0 = 10^3$. We choose this cell size, because it is the smallest one for which water depletion does not affect $t_{40}$. As far as $t_{40}$ is concerned, larger cells behave exactly like an ensemble of cells of this size. We find  $\langle t_{40}\rangle_{10^{-3}}/\langle t_{40}\rangle_1 \approx 3.7$ and $\langle t_{40}\rangle_{10^{-5}}/\langle t_{40}\rangle_1 \approx 6$. The value of  $\langle t_{40}\rangle_{10^{-5}}$ was estimated at 1090 s based on values along the diagonal for larger $\gamma$ and larger $N_0$. These ratios are lower than given in  Kostinski and Shaw (2005), showing that their theoretical analysis  slightly overestimates the "luckiness" in droplet growth. ~~Nevertheless, we agree with their conclusion that fluctuations play an important role in rain formation. Thanks to lucky collisions in some realizations (or, alternatively, in some parts of the cloud), mean concentration of rain drops after 30 minutes is about 200 m$^{-3}$. On the other hand, using the Smoluchowski equation leads to higher rain drop concentration than can be produced by lucky collisions (cf. Fig. 9). Significant role of fluctuations can also be seen in Fig. 8. Relative standard deviation of $\theta$ is high in small cells ($N_0 \leq 10^4$). This implies that small parts of the cloud could produce significant amount of rain much faster than average.~~

**7  Size of a well-mixed coalescence**

[revised manuscript text omitted]

---

## Referee Report (RR1)

**Review of "*Stochastic coalescence in Lagrangian cloud microphysics?*" by Dziekan and Pawlowska, Review of the first revision**

I appreciate very much the improvements made by the authors. The scientific content of the paper is novel and original. Nevertheless I believe that the presentation should be still improved in order to help future readers and to make sure it draws the attention it deserves. I still have the feeling that your presentation should be more explicit and that you should guide the readers better through the paper.

**Main points**

The newly added part towards the end of section 1 is a first step, but each section should start with a motivation and explain the goals behind the next tests. This is already nicely done in Section 4, 5, 6 and 7. On some occasions it may suffice to simply change the order of the presented material.

Some examples:

1. Whereas the intention of showing the size distribution in Fig.1 is clear to everyone and needs no further explanation, most readers probably do not know beforehand why you analyse of sigma(m_max)/E(m_max) and that this quantity is related to the gelation time. Your sentence on page 5, line 30 is essential in motivating what you do and hence should appear earlier.
2. You may still expand the description of the algorithm to make your paper self-contained such that readers are not forced to read Shima. Your whole paper is based on this algorithm, so I think it is worth investing a few more lines. It probably suffices to present a condensed version of their Eqs. 12-19, better explain multiple collections, and stress the point that only integer multiplicities are allowed. It may also help to highlight that the constant $N_{SD}$ simulation with $N_{SD}=N_0$ is different to a one-to-one simulation with $N_0$.
   Still I think that some formulations are not explicit enough. See e.g. my comment from the last review "you should write that one SD is created"… Contrary to your response, I do not think you already say it. You just say that the radius is randomly selected. Better write that "one SIP per bin is created…".
3. The following comment nicely demonstrates your "implicit" style of writing: The number of used realisations must be mentioned in the text, not only in the figure caption. In section 3, even the fact you analyse a certain number of realisations is not really mentioned directly. It is only implicitly clear by saying "average" or because you analyse sigmas.
4. Could you expand the description of Table 1 in the text? I do not understand the meaning of the third column. Hence, I am not able to reconstruct all parameters of the individual cases. In particular I'm confused about values like 2, 3, 200 etc.

**Discussion points**

I was wrong in stating that all SD methods solve the KCE (see points 4 and 5 of original review) and you convinced me that the probabilistic nature of the all-or-nothing approach has the same source of variability as the master equation. However, this is not true for the Riechelmann and Andrejczuk algorithms. They are not probabilistic (no Monte-Carlo approach is used, instead they solve the average KCE) and behave differently in the limiting case (explanation follows in the next paragraph).

So I would put it the following way: Some SD methods are based on KCE, some are based on the master equation. Hence, to avoid confusion, your statements throughout the manuscript should be reformulated, as they do not hold for SD methods in general.

In your and Shima's application of the all or nothing algorithm the multiplicities are integer values. Due to the design of the algorithm multiplicities remain integer, if integer values are used at initialisation. This is different in the Riechelmann and Andrejczuk algorithms, they produce real numbers, even for "integer initialisations". So the limiting case of a "one-to-one" simulation does not reduce to the master equation. Hence, my impression is that the various SD methods are not equivalent in the limiting case. Note that the all-or-nothing algorithm can also be applied with real numbers (see Unterstrasser et al, 2017.

The finding in Sec 4 "$N\_SD > 1/9 N\_0$" has strong implications on the feasibility of LES. May it be possible that with a full sampling of the SD pairs the constraint on $N\_SD$ is less strict? You may add at least a "full sampling" line for $N\_SD=32$ in Fig.5 to get a rough tendency. I acknowledge the tests you show in Figs.1&2, but those may not suffice to "prove" the equivalence between the full and linear sampling for all applications.

**Minor Points**

You define SSA as the algorithm by Gillespie, but later on SSA refers to the algorithm by Alfonso, doesn't it?

p.2., l.24: You miss to cite the Lagrangian cloud model by Sölch & Kärcher,2010. By the way it uses also the "all-or-nothing" approach for particle collisions.

p.3, l.20: I would say that only the opposite direction is true, i.e. all droplets in a SD are identical, but not all identical droplets are necessarily represented by one SD. You can well have two or ten SDs that represent all those droplets with similar properties.

p.4, l.24: you may add "which can only happen if two SDs with identical eta collide".

p.6, l.7: you probably refer to the **exponential** distribution used in Section 3. Check also for other occurences.

To make the connection between Fig. 5 and 6 clearer, you may use the same colours for the squares in Fig 6 as in Fig.5

p.8, l.1: simulational = computational?

p.9, l.6: How can it be that DNS results agree well with KCE results, even though in Fig. 4 you argue that the autoconversion time of KCE (and SDM) is too short?

p.10, l.13: I thought the conclusion would be that rain production is overestimated. What do you mean with amount of rain, mass or number of rain drops? Maybe it is the case, that the mass is overestimated and the number is underestimated?

p.12, l.3: Do you want to say "..it takes until the first droplet grows to r=40um"? Your formulation could cause confusion, as it changes over time, which droplet is the largest.

p.13,l14 and l15: 1.) the ratios must be flipped to get numbers >1.

2.) Did Kostinski & Shaw results depend on system size? Your results do: For N_0=10^4 and 10^5, the ratio goes down from 3.7 to 2.9 (=3912/1336) and 2.3 (=2552/1090). Whats the interpretation of this?

**Typos**

p.5, l.15: of discrepancies

p.6, l.12: each simulation -> any Bott simulation

ensemble, not ensamble

The reference list contains several small errors. I guess this is mostly due to the fact that in your bib file the paper titles are not embraced by {{ title}}. Then all words appear in lower case, see Alfonso & Raga, Li, Malinowski, ..

Unterstrasser should be cited with the GMD, not the GMDD article.

---

## Author Response (AR2)

**Response to the reviews**

September 20, 2017

We would like to express our gratitude to the reviewers for the time they invested in improving the paper. We have made changes to the manuscript in order to make the presentation more clear. Also, following the editor's advice, we have consulted a researcher from outside of our field on readability of the paper.

**Answer to the Anonymous Referee #2.**

**I appreciate very much the improvements made by the authors. The scientific content of the paper is novel and original. Nevertheless I believe that the presentation should be still improved in order to help future readers and to make sure it draws the attention it deserves. I still have the feeling that your presentation should be more explicit and that you should guide the readers better through the paper.**

**Main points**

**The newly added part towards the end of section 1 is a first step, but each section should start with a motivation and explain the goals behind the next tests. This is already nicely done in Section 4, 5, 6 and 7. On some occasions it may suffice to simply change the order of the presented material.**

Sections 2 and 3 now start with short introductions that explain their goals.

**Some examples: 1. Whereas the intention of showing the size distribution in Fig.1 is clear to everyone and needs no further explanation, most readers probably do not know beforehand why you analyse of sigma(m_max)/E(m_max) and that this quantity is related to the gelation time. Your sentence on page 5, line 30 is essential in motivating what you do and hence should appear earlier.**

The sentence now appears before we discuss Fig.2.

**2. You may still expand the description of the algorithm to make your paper self-contained such that readers are not forced to read Shima. Your whole paper is based on this algorithm, so I think it is worth investing a few more lines. It probably suffices to present a condensed version of their Eqs. 12-19, better explain multiple collections, and stress the point that only integer multiplicities are allowed. It may also help to highlight that the constant N_SD simulation with N_SD=N_0 is different to a one-to-one simulation with N_0. Still I think that some formulations are not explicit enough. See e.g. my comment from the last review you should write that one SD is created... Contrary to your response, I do not think you already say it. You just say that the radius is randomly selected. Better write that one SIP per bin is created....**

We agree that it will be easier for the readers if they do not have to go through all of the Shima's paper before reading our work. Section 2 has been considerably expanded following the suggestions. We believe that it now includes all necessary information on how coalescence is done in the super-droplet method.

**3. The following comment nicely demonstrates your implicit style of writing: The number of used realizations must be mentioned in the text, not only in the figure caption. In section 3, even the fact you analyse a certain number of realizations is not really mentioned directly. It is only implicitly clear by saying average or because you analyse sigmas.**

It is true that it may have been stated more clearly. We have made an effort to give all necessary information more explicitly throughout the paper.

**4. Could you expand the description of Table 1 in the text? I do not understand the meaning of the third column. Hence, I am not able to reconstruct all parameters of the individual cases. In particular Im confused about values like 2, 3, 200 etc.**

We have expanded the caption of Table 1.

**Discussion points**

**I was wrong in stating that all SD methods solve the KCE (see points 4 and 5 of original review) and you convinced me that the probabilistic nature of the all-or-nothing approach has the same source of variability as the master equation. However, this is not true for the**

**Riechelmann and Andrejczuk algorithms. They are not probabilistic (no Monte-Carlo approach is used, instead they solve the average KCE) and behave differently in the limiting case (explanation follows in the next paragraph).So I would put it the following way: Some SD methods are based on KCE, some are based on the master equation. Hence, to avoid confusion, your statements throughout the manuscript should be reformulated, as they do not hold for SD methods in general.**

We agree that some SD methods are based on the KCE. It was not our intention to give any conclusions on SD methods in general, only on the method of Shima. The misconception might be caused by the fact that there is no standard nomenclature in use. To clarify the text, now in the introduction we explicitly say that by the SD method we mean the method of Shima and that our conclusions are valid only for this method. In addition, we added a reference to the Shima method in the abstract.

**In your and Shimas application of the all or nothing algorithm the multiplicities are integer values. Due to the design of the algorithm multiplicities remain integer, if integer values are used at initialisation. This is different in the Riechelmann and Andrejczuk algorithms, they produce real numbers, even for integer initialisations. So the limiting case of a one-to-one simulation does not reduce to the master equation. Hence, my impression is that the various SD methods are not equivalent in the limiting case. Note that the all-or-nothing algorithm can also be applied with real numbers (see Unterstrasser et al, 2017.**

**The finding in Sec 4 N_SD > 1/9 N_0 has strong implications on the feasibility of LES. May it be possible that with a full sampling of the SD pairs the constraint on N_SD is less strict? You may add at least a full sampling line for N_SD=32 in Fig.5 to get a rough tendency. I acknowledge the tests you show in Figs.1&2, but those may not suffice to prove the equivalence between the full and linear sampling for all applications.**

It is true that LES with N_SD > 1/9 N_0 is not feasible. Using less SDs results in high fluctuations, but it does not mean that such simulations are useless. We have done LES of a stratocumulus with N_SD=100 and the resulting time series and averaged profiles are in qualitative agreement with results of LES with the KCE.

As suggested by the Reviewer, we have added a line for N_SD=32 with full sampling to Fig. 5. It is in good agreement with the line for N_SD=32 with linear sampling, so we conclude that the linear sampling does not affect the constraint on the number of SDs.

**Minor Points:**

**You define SSA as the algorithm by Gillespie, but later on SSA refers to the algorithm by Alfonso, doesnt it?**

SSA is the algorithm by Gillespie and it was used by Alfonso in the paper in which he introduces his own method. We now say it more clearly in Section 3.

**p.2., l.24: You miss to cite the Lagrangian cloud model by Slch & Krcher,2010. By the way it uses also the all-or-nothing approach for particle collisions.**

We added the citation.

**p.3, l.20: I would say that only the opposite direction is true, i.e. all droplets in a SD are identical, but not all identical droplets are necessarily represented by one SD. You can well have two or ten SDs that represent all those droplets with similar properties.**

We agree. The sentence now says that a SD represents *many*, not *all*, droplets with similar properties.

**p.4, l.24: you may add which can only happen if two SDs with identical eta collide.**

Done.

**p.6, l.7: you probably refer to the exponential distribution used in Section 3. Check also for other occurences.**

Yes, we refer to the distribution which is used in Section 3 to initialize droplet sizes. We think that it is clearly stated. We write that the *initial* droplet distribution is the same to avoid any misunderstanding that the droplet distribution after the simulation is the same as in Section 3.

**To make the connection between Fig. 5 and 6 clearer, you may use the same colours for the squares**

We did so.

**in Fig 6 as in Fig.5 p.8, l.1: simulational = computational?**

Changed.

**p.9, l.6: How can it be that DNS results agree well with KCE results, even though in Fig. 4 you argue that the autoconversion time of KCE (and SDM) is too short?**

That is a good point. The KCE results agree with DNS results only if the Long kernel is used in the KCE, what is now stated in Section 5. If the Hall kernel is used, the autoconversion time of the KCE is too short. We conclude that it is not the KCE approach that is invalid, but the kernel used.

**p.10, l.13: I thought the conclusion would be that rain production is overestimated. What do you mean with amount of rain, mass or number of rain drops? Maybe it is the case, that the mass is overestimated and the number is underestimated?**

We meant that the mass of rain is underestimated. Now we say explicitly that both the number of rain drops and the mass of rain are underestimated.

**p.12, l.3: Do you want to say ..it takes until the first droplet grows to r=40um? Your formulation could cause confusion, as it changes over time, which droplet is the largest.**

We have changed that following the suggestion.

**p.13,l14 and l15: 1.) the ratios must be flipped to get numbers ¿1.**

The error has been corrected.

**2.) Did Kostinski & Shaw results depend on system size? Your results do: For N_0=$10^4$ and $10^5$, the ratio goes down from 3.7 to 2.9 (=3912/1336) and 2.3 (=2552/1090). Whats the interpretation of this?**

That is a very good point and led us to change our conclusions - now we conclude that our results are in agreement with the Kostinski & Shaw theory. By calculating the ratios of t_40 we want to find out how much faster the luckiest droplet can grow compared to an average droplet. From the fact that values of t_40 are similar along the diagonals of Table 1 we concluded that, when looking for the time for the first droplet to grow to r=40 microns, large cells behave like ensembles of cells with N_0=$10^3$. We used to think that smaller cells distort the result, so we were calculating the "luck factor", i.e. the ratio of lucky t_40 to average t_40, for cells with N_0=$10^3$. The values of "luck factor" calculated using results for larger cells are smaller, because each larger cell itself is an ensemble of smaller cells, so it is an ensemble of droplets that can independently grow to r=40 microns.

The text and our reasoning has been changed in the following way. To better show that larger cells behave like an ensemble of smaller cells, we added a plot of $< t\_40 >$ vs $N\_0$ / gamma (Fig. 10). Based on this plot we conclude that even smaller cells, with $N\_0=10^2$, do not distort the results. Such cells can produce only a single drop with r=40 microns, so calculating the "luck factor" using these cells does in fact mean comparing single droplets and not ensembles of droplets. We find that the "luck factor" calculated from results for $N\_0=10^2$ is in agreement with the Kostinski & Shaw theory, so we conclude that their theory is correct.

**Typos**

**p.5, l.15: of discrepancies**
**p.6, l.12: each simulation -> any Bott simulation**
**ensemble, not ensamble**
**The reference list contains several small errors. I guess this is mostly due to the fact that in your bib file the paper titles are not embraced by title. Then all words appear in lower case, see Alfonso & Raga, Li, Malinowski, ..**
**Unterstrasser should be cited with the GMD, not the GMDD article.**

Typos have been fixed.

[revised manuscript text omitted]